# Cell-specific gain modulation by synaptically released zinc in cortical circuits of audition

Charles T Anderson[1†‡*], Manoj Kumar[1†], Shanshan Xiong[1,2],
Thanos Tzounopoulos[1*]

[1]Department of Otolaryngology, University of Pittsburgh, Pittsburgh, United States;
[2]The Third Xiangya Hospital, Central South University, Changsha, China

**Abstract** In many excitatory synapses, mobile zinc is found within glutamatergic vesicles and is coreleased with glutamate. Ex vivo studies established that synaptically released (synaptic) zinc inhibits excitatory neurotransmission at lower frequencies of synaptic activity but enhances steady state synaptic responses during higher frequencies of activity. However, it remains unknown how synaptic zinc affects neuronal processing in vivo. Here, we imaged the sound-evoked neuronal activity of the primary auditory cortex in awake mice. We discovered that synaptic zinc enhanced the gain of sound-evoked responses in CaMKII-expressing principal neurons, but it reduced the gain of parvalbumin- and somatostatin-expressing interneurons. This modulation was sound intensity-dependent and, in part, NMDA receptor-independent. By establishing a previously unknown link between synaptic zinc and gain control of auditory cortical processing, our findings advance understanding about cortical synaptic mechanisms and create a new framework for approaching and interpreting the role of the auditory cortex in sound processing.
DOI: https://doi.org/10.7554/eLife.29893.001

**\*For correspondence:**
charles.anderson@hsc.wvu.edu
(CTA);
thanos@pitt.edu (TT)

[†]These authors contributed
equally to this work

**Present address:** [‡]Department
of Physiology, Pharmacology,
and Neuroscience, Blanchette
Rockefeller Neurosciences
Institute, Sensory Neuroscience
Research Center, West Virginia
University School of Medicine,
Morgantown, United States

**Competing interests:** The
authors declare that no
competing interests exist.

**Reviewing editor:** Andrew J
King, University of Oxford,
United Kingdom

## Introduction

As a constituent for nearly 3000 proteins, zinc plays a key role in protein structure, enzymatic catalysis, and cellular regulation (**Vallee, 1988**). Although the chemistry and biology of zinc metalloproteins have historically dominated the field of zinc biology, there is a growing appreciation for a signaling role of free, mobile zinc found in secretory tissues such as the prostate, pancreas, and especially the brain (**Frederickson et al., 2005**; **Kelleher et al., 2011**). In many brain areas, including the neocortex, limbic structures and auditory brainstem, high concentrations of synaptic zinc are found in glutamatergic excitatory presynaptic vesicles (**Frederickson et al., 2005**). In the hippocampus, more than half of presynaptic glutamatergic contain synaptic zinc (**Sindreu et al., 2003**), attesting to zinc's abundance and importance in synaptic function.

Ex vivo studies established that synaptic zinc is an inhibitory neuromodulator of AMPA receptors, NMDA receptors, and vesicular release probability at lower frequencies of synaptic activity (**Kalappa et al., 2015**; **Anderson et al., 2015**; **Vergnano et al., 2014**; **Pan et al., 2011**; **Vogt et al., 2000**; **Perez-Rosello et al., 2013**). In contrast, during higher frequencies of synaptic activity and during enhanced vesicular release probability, synaptic zinc inhibits synaptic responses during the first few stimuli but enhances steady state responses during subsequent stimuli (**Kalappa and Tzounopoulos, 2017**). However, it is unknown whether and how zinc affects neuronal processing in vivo.

To answer this question, we investigated the role of synaptic zinc in shaping the sound-evoked activity in cortical neurons of awake mice. Namely, we investigated whether synaptic zinc modulates the relationship between sound intensity and the amplitude of neuronal evoked responses – termed gain modulation. Gain modulation controls the dynamic range of neuronal responses to sounds and

**eLife digest** Many people find it easy to follow a conversation while on a busy city street, but this seemingly simple task requires sophisticated processing of sounds. The brain must accurately distinguish speech sounds from background noise, even though the volumes and pitches of those sounds overlap. To make this possible, neurons that process sounds continually adjust the relationship between the volume of a sound and the size of their response. This helps the brain to distinguish more precisely between different sounds, but how this works remains unclear.

Zinc ions form part of almost 3,000 different enzymes and regulatory proteins, and also help neurons to communicate with one another at junctions called synapses. Changes to the amount of zinc ions at the synapses have been seen in disorders including depression and Alzheimer's disease. By imaging the brains of mice, Anderson, Kumar et al. now show that zinc ions affect how the healthy brain processes sounds.

Treating the mice with a substance that temporarily mops up zinc ions changed how neurons responded to sounds of different volumes. This revealed that zinc ions cause excitatory neurons, which activate neighboring cells, to increase their responses to sounds. Conversely, zinc ions cause inhibitory neurons, which reduce the activity of other cells, to decrease their responses to sounds. The overall effect is to change the balance of excitatory and inhibitory activity in areas of the brain that process sound. Anderson, Kumar et al. propose that these changes make it easier for the brain to process and distinguish different sounds as the environment changes from quiet to loud and vice versa.

As well as revealing a role for zinc ions in normal hearing, these findings may help us to understand disorders such as tinnitus and auditory neuropathies (conditions where the nerve that carries signals from the ear to the brain is damaged, leading to hearing loss). Both tinnitus and auditory neuropathies involve changes in the brain's ability to increase or decrease its responses to sounds with particular characteristics – processes that may involve the activity of zinc ions.

DOI: https://doi.org/10.7554/eLife.29893.002

is a fundamental feature of sound processing enabling adaptation to changes in sound stimulus statistics, as well as central compensation to peripheral damage (*Wen et al., 2012*; *Dean et al., 2005*; *Watkins and Barbour, 2008*; *Rabinowitz et al., 2011*; *Natan et al., 2017*; *Chambers et al., 2016*). We used widefield transcranial imaging of the genetically-encoded calcium indicator GCaMP6 to identify the effects of synaptic zinc on populations of specific neuronal types in the auditory cortex, and two-photon imaging to interrogate the effects of zinc on individual layer 2/3 neurons. Our results highlight synaptic zinc as a novel modulator of cortical responses to sound.

## Results

We began our exploration by visualizing sound-evoked responses in the auditory cortex of awake mice. To image and quantify the cortical sound-evoked responses, we used adeno-associated virus (AAV) driven by the synapsin promoter to express the genetically-encoded calcium indicator GCaMP6s in auditory cortical neurons (AAV-Syn-GCaMP6s, (*Chen et al., 2013*); Materials and methods). Eleven to 24 days following stereotaxic viral injections into the posterior temporal cortex, we performed in vivo calcium imaging in head-fixed unanesthetized mice (Materials and methods). To locate the primary auditory cortex (A1), we presented low frequency tones (5 or 6 kHz, 40–60 dB SPL) and imaged the sound-evoked changes in transcranial GCaMP6s fluorescence (*Figure 1a*, Materials and methods). Due to the mirror-like reversal of tonotopic gradients between A1 and the anterior auditory field (AAF) (*Guo et al., 2012*; *Joshi et al., 2015*), these sounds activated two discrete regions of the auditory cortex corresponding to the low frequency regions of A1 and the AAF (*Figure 1a* bottom). These results are consistent with previous studies mapping auditory cortical fields in the mouse (*Issa et al., 2014*; *Kato et al., 2015*; *Guo et al., 2012*).

After locating A1, we performed a small craniotomy adjacent to A1 and inserted a thin glass micropipette containing the extracellular, high-affinity, fast, zinc-specific chelator ZX1 (*Pan et al., 2011*; *Anderson et al., 2015*) (*Figure 1a* top right, Materials and methods). Then, we quantified the sound-evoked responses of A1 neuronal populations to 400 msec long, 12 kHz tones of 30–80 dB

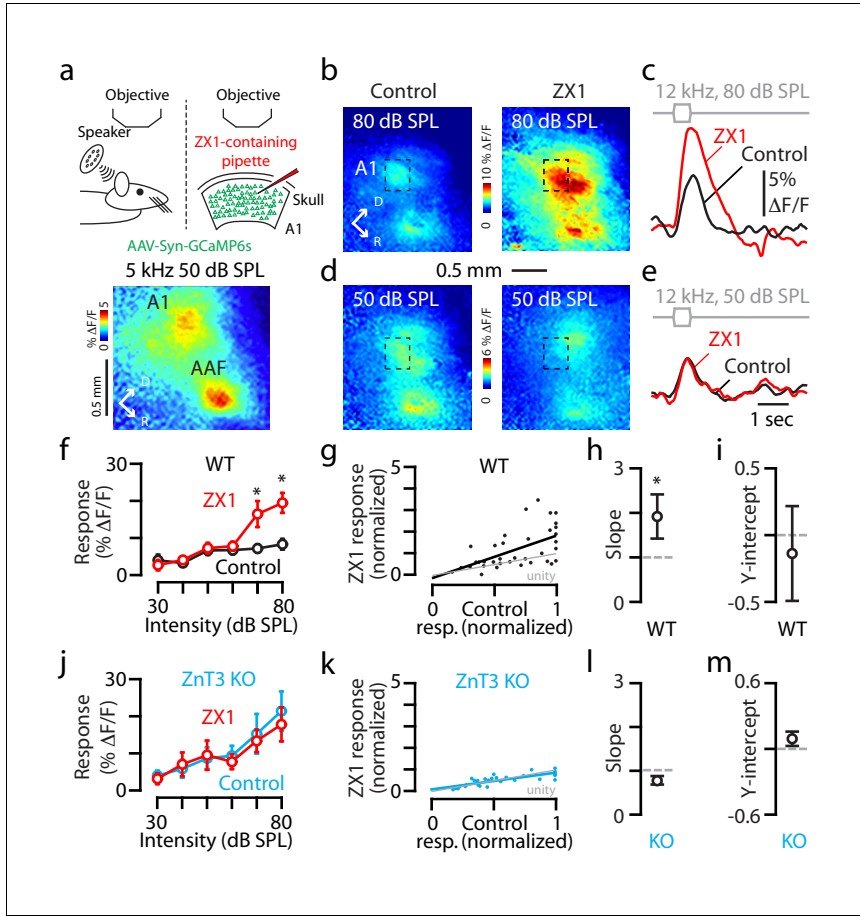

**Figure 1.** Synaptic, ZnT3-dependent zinc, decreases the gain of A1 sound-evoked responses in awake mice. (**a**) Schematic of experimental setup illustrating transcranial imaging using GCaMP6s in a head-fixed awake mouse (left panel). Sounds are delivered through a calibrated speaker (top left panel). A 5 kHz 50 dB SPL tone triggered GCaMP6s fluorescence responses in two regions of the auditory cortex representing A1 (bottom panel) and the anterior auditory field (AAF; D stands dorsal and R for rostral). ZX1-containing pipette was inserted into the cortex adjacent to A1 (top right panel). (**b–e**) A1 GCaMP6s fluorescence response to 12 kHz 80 or 50 dB SPL sounds before and after ZX1. (**f**) Average effect of control (black) and ZX1 (red), on A1 responses to 12 kHz sounds of different intensities (control vs. ZX1 for 70 dB: p=0.003; 80 dB: p=0.03, paired t-test, n = 7 mice). (**g**) Regression fit to normalized sound-evoked responses in control vs. ZX1 in WT; responses were normalized to the maximum response in control. (**h**) Regression slope (vs. 1, p=0.007, one sample t-test, n = 7 mice). (**i**) Regression y-intercept. (**j**) Same as in (**f**) but in ZnT3 KO (control is blue, n = 6 mice). (**k**) Same as in **g**) but in ZnT3 KO (n = 6 mice). (**l**) Same as in (**h**) but in ZnT3 KO (n = 6 mice). (**m**) Same as in (**i**) but in ZnT3 KO (n = 6 mice). For all the figures, asterisks indicate p<0.05 and error bars indicate SEM. For all the figures, see end of the manuscript for detailed values, Appendix 1.

DOI: https://doi.org/10.7554/eLife.29893.003

The following figure supplement is available for figure 1:

**Figure supplement 1.** ZX1 diffusion throughout the auditory cortex; ZnT3 KO mice show enhanced gain.

DOI: https://doi.org/10.7554/eLife.29893.004

SPL – an intensity span that encompasses sounds ranging from a quiet room to a busy city street. To assess the effect of zinc on A1 responses, we infused ZX1 in the auditory cortex and verified the intracortical diffusion of the chelator into A1 by visualizing the spread of the extracellular red fluorescent dye Alexa-594, co-infused with ZX1 (*Figure 1—figure supplement 1a–c*). We observed that the response to an 80 dB SPL tone was significantly enhanced after ZX1 application (*Figure 1b,c*); whereas, the response to a 50 dB SPL tone was not affected (*Figure 1d,e*). Overall, ZX1, but not vehicle containing ACSF and Alexa-594, enhanced the amplitude of the fluorescence response for

sounds of $\geq$70 dB SPL (*Figure 1f* and *Figure 1—figure supplement 1d,e*). These results indicate that endogenous extracellular zinc inhibits the gain of sound-evoked A1 responses.

Does zinc scale by division or shift by subtraction the gain of sound-evoked A1 responses? To answer this question, we plotted the normalized evoked responses before vs. after ZX1 infusion. By fitting a regression line to this function, we found a slope steeper than unity and no change in the y-intercept (*Figure 1g–i*), indicating that extracellular zinc signaling exerts divisive gain control of sound-evoked A1 responses.

To track the origin of the extracellular zinc modulating A1 gain control, we performed similar experiments in ZnT3 KO mice, which lack the vesicular zinc transporter ZnT3 and synaptic zinc (*Cole et al., 1999*). Compared to WT mice, ZnT3 KO mice showed larger responses to sounds of 80 dB SPL (*Figure 1—figure supplement 1f*), suggesting that synaptic zinc inhibits the gain of A1 responses. Moreover, zinc chelation in KO mice had no effect on the gain of A1 responses (*Figure 1j–m*, Materials and methods), suggesting that, in WT mice, ZX1 enhances A1 gain via chelation of extracellular, ZnT3-dependent zinc. Consistent with the high selectivity of ZX1 over calcium and other biologically relevant metal ions such as magnesium (*Pan et al., 2011*), the lack of ZX1 effects on the sound-evoked responses in ZnT3 KO mice further validate that the observed effects of ZX1 on the sound evoked responses of WT mice are due to the chelation of synaptic zinc – and not to the potential non-specific effects of ZX1 on calcium or other metal ions. Finally, because extracellular tonic, not synaptically-evoked, zinc levels in brain slices are ZnT3-independent (*Anderson et al., 2015*), our results suggest that synaptically released zinc modulates the gain of sound-evoked responses in A1.

We next investigated how synaptic zinc affects the gain of populations of specific classes of auditory cortical neurons. We recorded transcranial sound-evoked responses from populations of principal excitatory neurons, by utilizing an AAV driven by the calcium/calmodulin-dependent protein kinase 2 (CaMKII) promoter to express GCaMP6f in principal neurons (*Figure 2a–c* and *Figure 2—figure supplement 1a*; AAV-CaMKII-GCaMP6f, [*Pakan et al., 2016*]). ZX1 reduced the amplitude of sound-evoked responses of principal neurons to sounds of 60–80 dB SPL (*Figure 2a,b*). Linear regression analysis showed that the ZX1-induced reduction in the gain was due to a slope shallower than unity (*Figure 2c*), indicating that extracellular zinc signaling exerts multiplicative gain control of the sound-evoked A1 responses in principal neurons. In contrast to the enhancing effects of ZX1 on responses to sounds of 70–80 dB SPL in all neurons (*Figure 1f*), the divisive effect of ZX1 on responses to sounds of 60–80 dB SPL in principal neurons suggests that the effects of synaptic zinc are intensity- and cell type-specific.

To track the origin of the extracellular zinc increasing the gain of principal neurons, we studied the effects of ZX1 in ZnT3 KO mice, which had been injected with AAV-CaMKII-GCaMP6f. ZX1 had no effect on the sound-evoked responses of principal neurons in ZnT3 KO mice, indicating that synaptic zinc increases the gain of principal neurons (*Figure 2—figure supplement 1b,c*).

Whereas synaptic zinc is, mostly, an inhibitor of glutamatergic neurotransmission (*Perez-Rosello et al., 2013*; *Vergnano et al., 2014*; *Anderson et al., 2015*; *Kalappa et al., 2015*), zinc chelation decreased the sound-evoked activity of principal neurons. To explain this result, we hypothesized that the effects of zinc chelation are circuit-dependent. Since the sound-evoked responses of principal neurons in A1 are suppressed by inhibitory inputs from parvalbumin-expressing interneurons (PV neurons; *Li et al., 2013*; *Seybold et al., 2015*; *Phillips and Hasenstaub, 2016*; *Resnik and Polley, 2017*), a ZX1-induced increase of the sound-evoked responses of PV neurons would be consistent with the ZX1-induced decrease of the sound-evoked responses observed in principal neurons.

To image and quantify the sound-evoked responses of PV neurons, we injected AAV expressing Cre-dependent GCaMP6f (AAV-Flex-GCaMP6f, Materials and methods) into the auditory cortex of PV-Cre mice (*Figure 2—figure supplement 1a*, Materials and methods). ZX1 increased the sound-evoked responses of PV neurons to all tested sound intensities (30–80 dB SPL, *Figure 2d,e*). Linear regression analysis showed that the ZX1-induced increase in response gain was due to a slope steeper than unity (*Figure 2f*), indicating that extracellular zinc exerts divisive gain control of sound-evoked responses in PV neurons.

In PV neurons, the enhancing effect of ZX1 on sound-evoked responses was largest for sounds of 60–80 dB SPL (*Figure 2—figure supplement 1d*) – the intensities where principal neurons showed a ZX1-induced reduction in their responses (*Figure 2b*). Although our approach involving the bulk

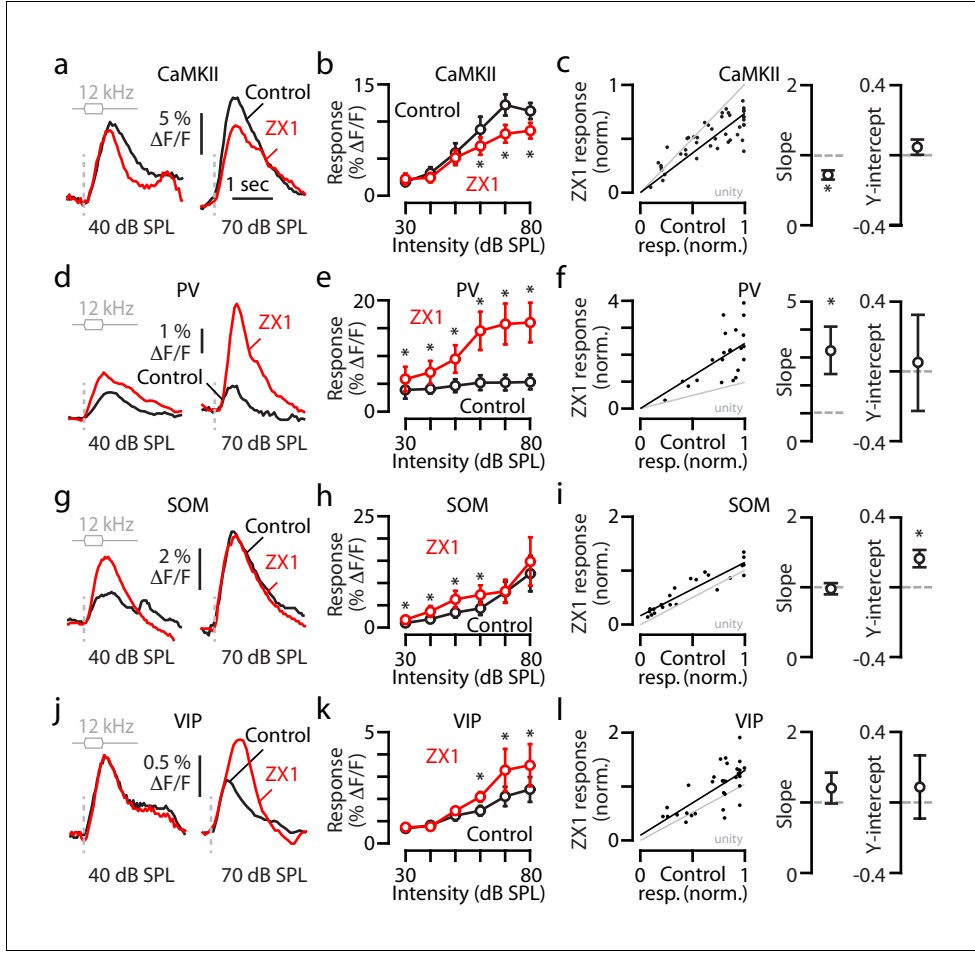

**Figure 2.** Synaptic zinc increases the gain of principal neurons but decreases the gain of PV and SOM interneurons. (a) Transcranial fluorescence responses of principal neurons to 12 kHz sounds of 40 and 70 dB SPL before and after ZX1. (b) Average effect of control (black) and ZX1 (red), on principal neuron responses to 12 kHz sounds of different intensities (control vs. ZX1 for 60 dB: p=0.02, 70 dB: p=0.001; 80 dB: p=0.003, n = 10 mice, paired t-tests). (c) Left: Regression fit to normalized sound-evoked responses in control vs. ZX1 in principal neurons. Right: The slope and y-intercept of the regression (regression slope vs. 1: p=0.001). (d) Same as in (a) but for PV neurons. (e) Same as in b) but for PV neurons (control vs. ZX1 for 30 dB: p=0.009, 40 dB: p=0.004, 50 dB: p=0.04, 60 dB: p=0.006, 70 dB: p=0.017; 80 dB: p=0.011, n = 4 mice, paired t-tests). (f) Same as in c) but for PV neurons (regression slope vs. 1: p=0.03). (g) Same as in (a) but for SOM neurons. (h) Same as in b) but for SOM neurons (control vs. ZX1 for 30 dB: p=0.01, 40 dB: p=0.04, 50 dB: p=0.03, 60 dB: p=0.03, n = 4 mice, paired t-tests). (i) Same as in (c) but for SOM neurons (regression y-intercept vs. 0: p=0.01). (j) Same as in (a) but for VIP neurons. (k) Same as in (b) but for VIP neurons (control vs. ZX1 for 60 dB: p=0.03, 70 dB: p=0.008; 80 dB: p=0.009, n = 7 mice, paired t-tests). (l) Same as in (c) but for VIP neurons (n = 7 mice).

DOI: https://doi.org/10.7554/eLife.29893.005

The following figure supplement is available for figure 2:

**Figure supplement 1.** Cell-specific expression of GCaMP6f in A1; ZnT3-dependence of gain effects on principal neurons; and intensity-dependent effects of ZX1.
DOI: https://doi.org/10.7554/eLife.29893.006

application of ZX1 is not conclusive for distinguishing cell- (direct) vs. circuit-dependent effects (see Discussion for details), this coupling between the ZX1-induced enhancement in the gain of PV neurons (*Figure 2ef*) and the ZX1-induced reduction in the gain of principal neurons (*Figure 2bc*) is consistent with the notion that synaptic zinc increases, at least in part, the gain of principal neurons by decreasing the gain of PV neurons.

We next focused on the effect of zinc chelation on the somatostatin-expressing interneurons (SOM neurons), which inhibit PV neurons (*Harris and Shepherd, 2015*; *Tremblay et al., 2016*). We selectively expressed GCaMP6f in SOM neurons by injecting AAV-Flex-GCaMP6f into SOM-Cre mice (*Figure 2—figure supplement 1a*, Methods). ZX1 increased the sound-evoked responses of SOM neurons to sounds of lower intensities (30–60 dB SPL, *Figure 2gh*, *Figure 2—figure supplement 1d*), and increased the gain of these neurons through a positive shift in the y-intercept (*Figure 2i*). These results support that extracellular zinc exerts subtractive gain control of sound-evoked A1 responses in SOM neurons.

The smaller ZX1-induced enhancement of the sound-evoked responses of PV neurons to lower intensity sounds (30–40 dB SPL) is consistent with the ZX1-induced increases of the sound-evoked responses of SOM neurons to the same sound intensities (*Figure 2—figure supplement 1d*). Together, our results from SOM, PV and principal neurons are consistent with a scheme where the zinc-mediated inhibition of the sound-evoked responses in SOM neurons contributes to the sound intensity dependence of the zinc-mediated inhibition in PV neurons, which, in turn, contributes to the zinc-mediated increase in the sound-evoked responses of principal neurons (see Discussion for more details).

We next investigated the effect of zinc chelation on the vasoactive intestinal polypeptide-expressing interneurons (VIP neurons), which inhibit SOM and PV neurons (*Harris and Shepherd, 2015*; *Tremblay et al., 2016*; *Pi et al., 2013*). We selectively expressed GCaMP6f in VIP neurons by injecting AAV-Flex-GCaMP6f into VIP-Cre mice (*Figure 2—figure supplement 1a*, Materials and methods). ZX1 increased the responses of VIP neurons to sounds of 60–80 dB SPL but did not affect their overall response gain (*Figure 2j–l*). The ZX1-induced enhancements of the responses of VIP neurons to sounds of 60–80 dB SPL are consistent with the lack of ZX1 effects on the responses of SOM neurons to sounds of the same intensity (*Figure 2—figure supplement 1d*). Although the lack of ZX1 effects on the gain of VIP neurons may reflect either a circuit-based cancellation of opposing effects or no direct effect (see Discussion for more details), together, our results show that synaptic zinc is a novel modulator of cortical sound processing – a modulator that increases the gain of principal neurons, but reduces the gain of PV and SOM neurons.

We next focused on the molecular targets (i.e., zinc-interacting proteins) that contribute to the effects of synaptic zinc on cortical sound-evoked activity. Because nanomolar levels of zinc inhibit NMDA receptors (NMDARs; *Paoletti et al., 1997*; *Hansen et al., 2014*), we tested whether NMDARs contribute to the observed zinc-mediated gain modulation in A1. To answer this question, we measured the effects of ZX1 on the neuronal sound-evoked responses after pharmacological blockade of NMDARs with a selective NMDAR antagonist (D-2-Amino-5-phosphonopentanoic acid, APV, infusion, Materials and methods). In the presence of APV (control), ZX1 increased the responses of principal neurons to sounds of 70–80 dB SPL (*Figure 3ab*), which is a reversal of the observed ZX1-induced reduction in the sound-evoked responses in the absence of APV (*Figure 2ab*). Moreover, APV enhanced the sound-evoked responses of principal neurons (*Figure 3—figure supplement 1ab*), suggesting that APV disinhibits principal neurons. This disinhibition may also disrupt the coupling between the ZX1-induced enhancement in the gain of PV neurons (*Figure 2e,f*) and the ZX1-induced reduction in the gain of principal neurons (*Figure 2b,c*). Consistent with this hypothesis, in the presence of APV, zinc chelation caused an increase in the sound-evoked responses of PV neurons to sound intensities of 30–80 dB SPL (*Figure 3a,c*). These results suggest that the disinhibition of principal neurons, in the presence of APV, unmasked the NMDAR-independent inhibitory effects of zinc on the sound-evoked responses of principal neurons, which are likely direct and non-circuit-dependent. However, similarly to ZX1 application, our approach involving the bulk application of APV is not conclusive for distinguishing cell autonomous vs. circuit-dependent effects (see Discussion for more details).

The ZX1-induced increases in the sound-evoked responses of PV neurons in the presence of APV (*Figure 3c*) were not different from the increases observed in the absence of APV (*Figure 2f*), suggesting that the effects of zinc on the responses of PV neurons are, at least in part, NMDAR-independent.

Next, we investigated whether NMDARs contribute to the effects of synaptic zinc on the sound-evoked responses of SOM neurons. APV reduced the sound-evoked responses of SOM neurons (*Figure 3—figure supplement 1ef*) and abolished the effects of zinc chelation on the sound-evoked responses of SOM neurons (*Figure 3d,e*). Although potential circuit effects due to NMDAR-

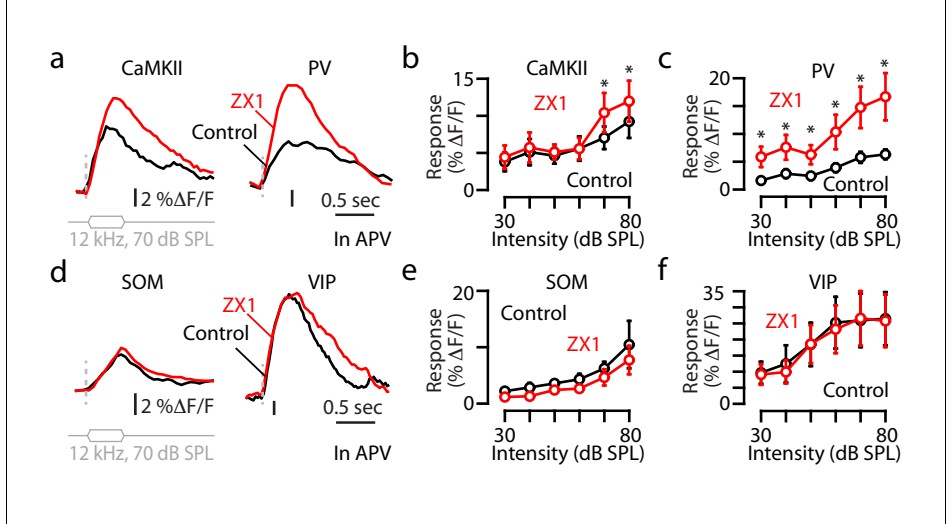

**Figure 3.** Non-NMDAR signaling mediates zinc inhibition of the sound-evoked responses of principal and PV neurons. (**a**) In the presence of APV, fluorescence responses of principal neurons (left) and PV neurons (right) to 12 kHz sounds of 70 dB SPL before and after ZX1. (**b**) In the presence of APV, average responses of principal neurons to 12 kHz sounds of different intensities in control (APV, black) and after ZX1 (red) (control vs. ZX1 for 70 dB: p=0.04; 80 dB: p=0.01, n = 6 mice, paired t-tests). (**c**) Same as in **b**) but for PV neurons (control vs. ZX1 for 30 dB: p=0.04, 40 dB: p=0.03, 50 dB: p=0.02, 60 dB: p=0.04, 70 dB: p=0.01; 80 dB: p=0.01, n = 7 mice, paired t-tests). (**d**) Same as in (**a**) but for SOM and VIP neurons. (**e**) Same as in (**b**) but for SOM neurons (n = 4 mice). (**f**) Same as in (**b**) but for VIP neurons (n = 4 mice).

DOI: https://doi.org/10.7554/eLife.29893.007

The following figure supplement is available for figure 3:

**Figure supplement 1.** The effect of APV on the sound-evoked responses of principal, PV, SOM and VIP neurons.

DOI: https://doi.org/10.7554/eLife.29893.008

dependent and NMDAR-independent effects that cancel each other out can't be ruled out (see Discussion for more details), our results are consistent with the notion that the effects of synaptic zinc on the sound-evoked responses of SOM neurons are NMDAR-dependent.

We next tested whether NMDARs contribute to the effects of synaptic zinc on the sound-evoked responses of VIP neurons. APV enhanced the sound-evoked responses of VIP neurons (*Figure 3— figure supplement 1gh*) and abolished the effects of ZX1 on the sound-evoked responses of VIP neurons (*Figure 3df*), suggesting that synaptic zinc inhibits the sound-evoked responses of VIP neurons, likely, via NMDAR-dependent mechanisms. However, similarly to the limitations in the interpretation of our results in SOM neurons due to potential circuit-based cancelation, NMDAR-independent mechanisms can't be excluded. Overall, our results highlight that synaptic zinc modulates non-NMDAR targets to inhibit the gain of sound-evoked responses of principal and PV neurons; and suggest that NMDAR-mediated signaling is involved in the zinc-mediated gain control of SOM and VIP neurons.

Transcranial imaging reflects a sound-evoked population fluorescent signal arising from neurons residing in different cortical layers, as well as from different neuronal compartments (e.g. somata vs. dendrites). To explore the layer and sub-cellular specificity of the zinc-mediated effects on gain control, we next performed two-photon imaging of individual A1 L2/3 neurons in awake mice. In these experiments, we investigated the effects of ZX1 on the sound evoked responses from individual L2/3 neuronal somata (Methods). We began these experiments by injecting AAV-CaMKII-GCaMP6f into A1 to record responses from principal neurons (Materials and methods). After locating A1 as shown in *Figure 1*, we performed a craniotomy and subsequent two-photon imaging to obtain sound-evoked responses from the somata of individual principal neurons (*Figure 4a*, Materials and methods). We then infused ZX1 and were able to locate the same group of neurons to remeasure their sound-evoked responses (*Figure 4a*). ZX1 reduced the responses to sounds of 70 dB SPL (*Figure 4b,c*), and linear regression analysis showed that this reduction in gain was divisive

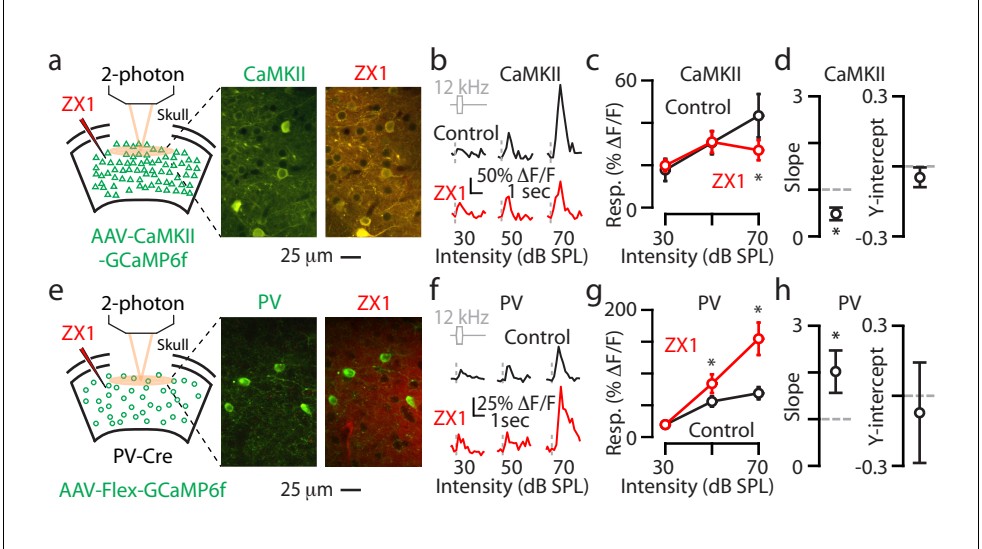

**Figure 4.** Synaptic zinc increases the gain of individual L2/3 principal neurons and decreases the gain of individual L2/3 PV neurons. (a) Left: Schematic of experimental setup illustrating 2-photon imaging of GCaMP6f in principal neurons. Middle: Image of a population of A1 L2/3 principal neurons. Right: The same neurons as in the middle panel after ZX1 infusion. (b) Representative example of the sound evoked responses from a L2/3 principal neuron in control (black) and after ZX1 infusion (red). (c) Average responses of principal neurons to 12 kHz sounds of different intensities in control (black) and after ZX1 (red) (control vs. ZX1, 70 dB: p=0.04, signed-rank test, n = 86 neurons from 5 mice). (d) Regression analysis of the sound-evoked responses in control vs. ZX1 in principal neurons. Left: linear regression slope (vs. 1, p=0.001, n = 86 neurons from 5 mice, one sample t-test). Right: linear regression y-intercept. (e) Same as in (a) but for PV neurons. (f) Same as in (b) but for PV neurons. (g) Same as in (c) but for PV neurons (ZX1 vs. control (black), 50 dB: p=0.02 70 dB: p=0.001, signed-rank test, n = 40 neurons from 7 mice. (h) Same as in (d) but for PV neurons (regression slope vs. 1: p=0.04, n = 40 neurons from 6 mice, one sample t-test).

DOI: https://doi.org/10.7554/eLife.29893.009

(*Figure 4d*), which is consistent with our results obtained with transcranial imaging (*Figure 2c*). These results show that synaptic zinc exerts multiplicative gain control of sound-evoked responses in individual L2/3 principal neurons.

We next measured the effects of ZX1 on the responses of individual L2/3 PV neurons. We injected AAV-Flex-GCaMP6f into PV-Cre mice and performed 2-photon imaging (*Figure 4e*). We measured the sound-evoked responses from the somata of individual L2/3 PV neurons, infused ZX1, and remeasured the responses of the same neurons (*Figure 4e*). PV neurons showed increased sound-evoked responses to sounds of 50 and 70 dB SPL (*Figure 4f,g*) and a multiplicative increase in their gain (*Figure 4h*), suggesting that synaptic zinc exerts divisive gain control of sound-evoked responses in individual L2/3 PV neurons.

In individual L2/3 PV neurons, in contrast to our transcranial measurements, we did not observe any increases in their somatic responses to sounds of 30 dB SPL. Since calcium transients in neuronal dendrites can be tuned to different stimulus features compared to somata (*Chen et al., 2013*), and can generate action potentials independently from the soma (*Golding and Spruston, 1998*), this discrepancy may reflect the differential zinc modulation of sound-evoked responses in dendritic vs. somatic domains of L2/3 PV neurons. It could also indicate that the sound-evoked responses of non-L2/3 PV neurons may contribute to the effects synaptic zinc to sounds of 30 dB SPL observed with transcranial imaging. Together, these results show that synaptic zinc signaling decreases the gain of sound-evoked responses of individual L2/3 PV neurons in A1.

We performed similar two-photon experiments in individual L2/3 SOM and VIP neurons (*Figure 5*) and the results we obtained were, overall, consistent with the results we observed with transcranial imaging: ZX1 caused an additive gain increase in L2/3 SOM neurons (*Figure 5a–d*) and no significant changes in the gain of L2/3 VIP neurons (*Figure 5e–h*). Whereas with transcranial imaging we

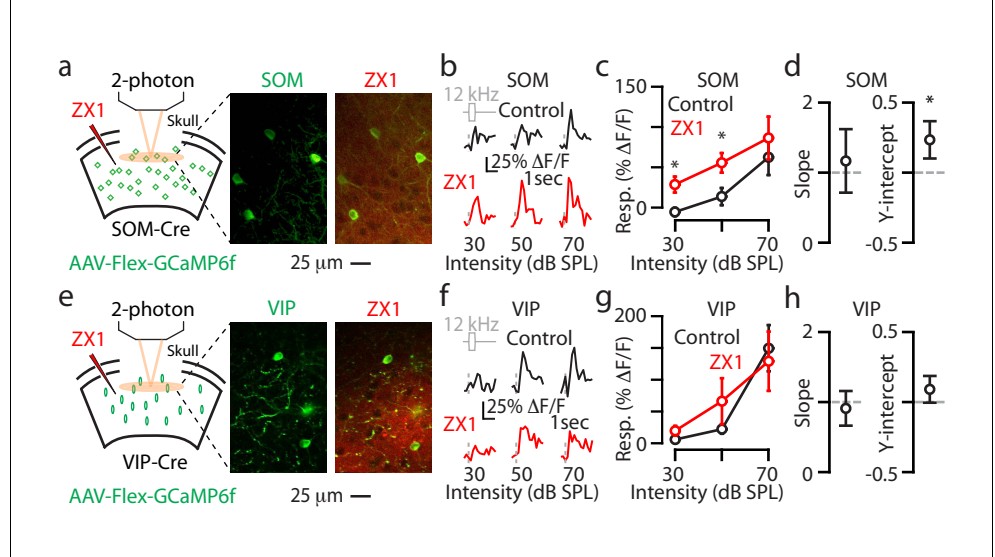

**Figure 5.** Synaptic zinc decreases the gain of individual L2/3 SOM neurons and does not affect the gain of individual L2/3 VIP neurons. (**a**) Left: Schematic of experimental setup illustrating 2-photon imaging of GCaMP6f in SOM neurons. Middle: Image of a population of A1 L2/3 SOM neurons. Right: The same neurons as in the middle panel after ZX1 infusion. (**b**) Representative example of the sound evoked responses from a L2/3 SOM neuron in control (black) and after ZX1 infusion (red). (**c**) Average responses of L2/3 SOM neurons to 12 kHz sounds of different intensities in control (black) and ZX1 (red) (control vs. ZX1 , 30 dB: p=0.02, 50 dB: p=0.003, signed-rank test, n = 12 neurons from 4 mice. (**d**) Regression analysis of the sound-evoked responses in control vs. ZX1 in SOM neurons. Left: linear regression slope. Right: linear regression y-intercept (vs. 0, p=0.001, n = 12 neurons from 4 mice, one sample t-test). (**e**) Same as in (**a**) but for VIP neurons. (**f**) Same as in (**b**) but for VIP neurons. (**g**) Same as in (**c**) but for VIP neurons (n = 30 neurons from 5 mice). (**h**) Same as in (**d**) but for VIP neurons (n = 30 neurons from 6 mice).

DOI: https://doi.org/10.7554/eLife.29893.010

observed ZX1-induced increases in the responses of VIP neurons to sounds of 70 dB SPL, we did not observe such an increase in individual L2/3 VIP neurons. This discrepancy may reflect the differential zinc modulation of sound-evoked responses in dendritic vs. somatic domains of L2/3 VIP neurons and/or may suggest that non-L2/3 VIP neurons are responsible for the enhancement we observed with transcranial imaging. Together, our two-photon experiments on individual L2/3 neurons support the cell-specific effect of synaptic zinc on A1 neurons: synaptic zinc increases the gain of principal neurons, but decreases the gain of PV and SOM interneurons.

## Discussion

Our results show that synaptic zinc modulates the gain of sound-evoked auditory cortical responses. In the auditory system, as sound statistics of the auditory environment change, auditory neurons adjust their gain – the dynamic range of their responses – to match the stimulus statistics. In the auditory nerve, inferior colliculus, and auditory cortex, neurons adapt their responses to match the mean sound intensity levels (*Wen et al., 2012*; *Dean et al., 2005*; *Watkins and Barbour, 2008*). In addition to adaptation to the mean sound intensity levels, auditory cortical response gain is modulated by higher-level stimulus statistics, such as spectrotemporal contrast, reflecting tone distributions with identical mean intensity but different widths, and stimulus temporal correlation (*Rabinowitz et al., 2011*; *Natan et al., 2017*). Moreover, increases in cortical gain restore sound-evoked responses following near complete cochlear denervation, thus providing central compensation to peripheral damage (*Chambers et al., 2016*). Therefore, gain modulation provides a mechanism for generating sensory representations that are robust to noise, thus maximizing sensitivity to changes in stimulus intensity across variable auditory contexts. Moreover, gain modulation maintains the robustness of central responses after peripheral damage. We propose, although not addressed

in this manuscript, that sound-dependent changes in zinc signaling may contribute to these mechanisms. But, is there any evidence for activity-dependent changes in cortical zinc signaling?

Sensory experience alters synaptic zinc levels in primary somatosensory and visual cortex (*Brown and Dyck, 2002*; *Dyck et al., 2003*); however, the physiological roles of this plasticity in sensory processing remain unknown. Based on our results, we propose that sound-dependent changes in zinc signaling, which have been demonstrated in the dorsal cochlear nucleus (*Kalappa et al., 2015*), may contribute to the activity-dependent changes in cortical gain. Such a contribution, which will be the focus of future experiments, would provide a role for zinc signaling in the adaptation of neuronal input-output functions to changing sound statistics, as well as in the preservation of central responses after peripheral damage.

Synaptic zinc is present in many cortical areas (*Danscher and Stoltenberg, 2005*) and dynamic gain modulation is a fundamental mechanism across many brain regions affecting sensory computations, attention, multisensory integration and value estimation (*Somers et al., 1995*; *Reynolds and Heeger, 2009*; *Ohshiro et al., 2011*; *Louie et al., 2013*). We therefore propose, although not addressed in this manuscript, that activity-dependent modification of synaptic zinc signaling may be a fundamental mechanism capable of mediating dynamic gain modulation in response to context, history, training, or injury across sensory modalities.

Gain modulation is mediated by selective increases or decreases in cortical inhibition (*Li et al., 2013*; *Olsen et al., 2012*; *Wilson et al., 2012*; *Phillips and Hasenstaub, 2016*). Recent studies, including optogenetic approaches to increase the activity of PV and SOM neurons, have suggested that PV neurons provide multiplicative/divisive gain control of principal neurons, whereas SOM neurons provide additive/subtractive gain control of principal neurons (*Atallah et al., 2012*; *Li et al., 2014*; *Wilson et al., 2012*; *Moore and Wehr, 2013*). However, activation and inactivation of PV and SOM neurons revealed asymmetric effects on neuronal gain in auditory cortex (*Phillips and Hasenstaub, 2016*). Here, we observed that zinc caused a multiplication in the gain of principal neurons, which is consistent with the observed zinc-mediated division in the gain of PV neurons. Therefore, our results are consistent with the role of PV neurons in providing dynamic gain control in A1 and add zinc as a player in the gain control of L2/3 cortical circuits.

The lack of ZX1 effects on the A1 neuronal responses to specific sound intensities may reflect circuit-based cancelation of opposing ZX1 effects. Although the bulk application of ZX1 limits our ability to distinguish cell- from circuit-specific effects, the lack of ZX1 effect on the sound-evoked responses of principal and VIP neurons to lower sound intensities (30–50 dB SPL), and on the responses of SOM neurons to higher intensities (70–80 dB SPL) suggests that the disruption of extracellular zinc signaling does not affect the balance of excitation and inhibition in these cases (*Wehr and Zador, 2003*). In principal neurons, it is interesting that this balance begins to break at sound intensities of $\geq$60 dB SPL, which is the intensity range where we observed the ZX1-induced increases in the sound-evoked responses of VIP neurons. Since VIP neurons provide inhibition to both SOM and PV neurons (*Pi et al., 2013*; *Harris and Shepherd, 2015*), these findings suggest that synaptic zinc modulates the VIP-mediated disinhibition of SOM and PV neurons, which, in turn, shape the gain of principal neurons. This scheme, and our previously stated suggestions on the hierarchical contribution of the zinc-mediated effects on SOM, PV and principal neuronal responses to sound (Results), are consistent with the overall sequential hodology of the three main inhibitory cortical cell classes (*Harris and Shepherd, 2015*). Importantly, the effects of ZX1 over a wide range of sound intensities (30–80 dB SPL) reflect the ethological relevance of synaptic zinc signaling in modulating cortical responses to sound.

Blockade of NMDARs with APV reduced the sound-evoked responses of PV neurons (*Figure 3—figure supplement 1cd*) and increased the sound-evoked responses of principal neurons (*Figure 3—figure supplement 1ab*). Similarly, in the prefrontal cortex pharmacological blockade of NMDARs reduced the spontaneous firing rates of PV neurons and increased the spontaneous firing rates of principal neurons (*Homayoun and Moghaddam, 2007*). Moreover, genetic deletion of NMDARs selectively from PV neurons increased the spontaneous firing rates of principal neurons (*Carlén et al., 2012*), suggesting that NMDA receptors expressed by PV-neurons contribute to PV neuron-mediated inhibition of principal neurons. Finally, PV neurons show high expression of GluN2A containing NMDAR subunits (*Xi et al., 2009*), which are exceptionally sensitive to low nanomolar zinc levels (*Paoletti et al., 1997*; *Hansen et al., 2014*), thus rendering excitatory inputs to PV neurons very zinc-sensitive. Together, these results are consistent with the view that NMDAR

signaling and its modulation by zinc support and modulate the ability of PV neurons to inhibit principal neurons. However, it is unclear whether NMDARs expressed by PV neurons are responsible for these effects, for previous studies showed relatively weak NMDA receptor-mediated excitation of PV neurons (*Goldberg et al., 2003*; *Rotaru et al., 2011*), suggesting that NMDARs are not a strong source of excitation for PV neurons. In this context, it is interesting that blockade of NMDARs increased the sound-evoked responses of VIP neurons (*Figure 3—figure supplement 1gh*), which provide inhibition to PV neurons (*Tremblay et al., 2016*). This suggests that NMDAR signaling may serve to increase the sound evoked-responses of PV neurons via a combination of direct excitation of PV neurons and reduced VIP-mediated sound-evoked inhibition to these cells. Future studies will be required to explore these possibilities.

The responses of PV neurons to sound are the most sensitive to zinc modulation, both in amplitude and sound intensity span (*Figure 2* and *Figure 2—figure supplement 1d*). It is interesting to note that plasticity in auditory cortex PV neurons during the first days following auditory nerve damage predicts the eventual recovery of cortical sound processing, which is achieved via increased A1 gain modulation weeks later (*Resnik and Polley, 2017*). The mechanisms of this PV neuronal plasticity remain unknown; however, activity-dependent changes in zinc levels may contribute to this plasticity – future experiments will be needed to test this hypothesis.

We observed both NMDAR-dependent and NMDAR-independent zinc-mediated effects on A1 responses to sound. This is consistent with in vitro studies showing that synaptic zinc inhibits, albeit at different concentrations, NMDARs (*Paoletti et al., 1997*; *Hansen et al., 2014*), AMPA receptors (*Kalappa et al., 2015*; *Kalappa and Tzounopoulos, 2017*), and reduces vesicular release probability via endocannabinoid signaling (*Perez-Rosello et al., 2013*; *Kalappa and Tzounopoulos, 2017*). Although previous studies have reported potentiation, inhibition and no effects of zinc on AMPARs (*McAllister and Dyck, 2017*), the vast majority of these did not employ ZX1, which, due to its improved kinetic and zinc binding properties, is the most appropriate chelator for investigating the effects of fast, transient elevations of synaptic zinc on synaptic targets (*Pan et al., 2011*; *Anderson et al., 2015*; *Kalappa et al., 2015*). However, ZX1 had no effects on synaptic AMPA currents in mossy fiber-CA3 synapses, suggesting that the effects of zinc on synaptic AMPA responses are subunit- and/or synapse-specific. Together, our results suggest that synaptic zinc shapes the gain of L2/3 neurons via multiple zinc-interacting targets, including NMDARs.

In the presence of APV, the lack of ZX1 effects on the sound-evoked responses of SOM and VIP neurons may be due to potential circuit- or to direct but multiple target-based cancelation of ZX1 effects, and it therefore does not necessarily support total NMDAR-dependence. Despite these limitations in the interpretation of the APV experiments showing negative results, our APV experiments showing enhancing effects of ZX1 on the sound-evoked responses of principal and PV neurons, suggest that synaptic zinc exerts its effects on these neurons, at least partially, in an NMDAR-independent manner. Nonetheless, future experiments investigating the effect of sound-evoked zinc on subthreshold conductances mediating the sound-evoked response of different types of neurons in A1; in vitro experiments on the effect of synaptic zinc on NMDAR and AMPAR EPSCs in A1 synapses; as well as the use of optogenetic approaches are needed to further resolve the detailed molecular mechanisms via which synaptic zinc shapes the sound-evoked responses of A1 L2/3 neurons.

Our findings are mostly specific to cells and circuits in L2/3. In the neocortex, synaptic zinc containing terminals are found predominantly in L1, 2/3 and 5, with moderate presence in L6 and light presence in L4 (*McAllister and Dyck, 2017*). This distribution together with the differential distribution of the different classes of interneurons – with VIP neurons more predominant in the superficial layers, SOM neurons more predominant in the deeper layers, and PV neurons present throughout the cortex (*Tremblay et al., 2016*) – make it unclear whether our findings apply to the deeper layers of cortex. Future studies will be required to address the effects of synaptic zinc on the gain of neurons and circuits located outside of L2/3.

Together, our results and previous findings on the role of zinc in modulating glutamatergic neurotransmission demonstrate that synaptic zinc fine-tunes excitatory synaptic transmission and sensory processing. Although this fine-tuning does not impair baseline neurotransmission or basic sensory and sensorimotor functions, such as hearing thresholds and prepulse inhibition (*Cole et al., 2001*; *Thackray et al., 2017*), it has profound consequences on synaptic plasticity (*Pan et al., 2011*), sound-evoked cortical processing and more intricate, yet ethologically crucial sensory processing

tasks, such as gain modulation. We propose that zinc is a novel 'knob' in the brain that tunes cortical gain.

## Materials and methods

### Animals

All procedures were approved by the Institutional Animal Care and Use Committee at the University of Pittsburgh. Male and female ZnT3 (*Slc30a3*) KO and WT mice (Jackson, Bar Harbor, ME), were used for experiments shown in *Figure 1a–m*, *Figure 1—figure supplement 1a–c,f*, and *Figure 2—figure supplement 1bc*. Male and female ICR mice (Harlan, Indianapolis, IN) were used for experiments shown in *Figure 2a–c*, *Figure 3a,b*, *Figure 4a–d*, *Figure 1—figure supplement 1a–f*, *Figure 2—figure supplement 1a,d* (AAV-CaMKII-GCaMP6f), and *Figure 3—figure supplement 1ab*. PV(*Pvalb*)-Cre mice (Jackson, Bar Harbor, ME) were used for experiments shown in *Figure 2d–f*, *Figure 3a–c*, *Figure 4e–h*, *Figure 2—figure supplement 1a–d*, and *Figure 3—figure supplement 1cd*. SOM(*Sst*)-Cre mice (Jackson, Bar Harbor, ME) were used for experiments shown in *Figure 2g–i*, *Figure 3d,e*, *Figure 5a–d*, *Figure 2—figure supplement 1a–d*, and *Figure 3—figure supplement 1ef*. VIP-Cre mice (Jackson, Bar Harbor, ME) were used for experiments shown in *Figure 2j–l*, *Figure 3d,f*, *Figure 5e–h*, *Figure 2—figure supplement 1a,d*, and *Figure 3—figure supplement 1gh*.

### AAV injections

Mice between postnatal day (P) 19 and P36 were anesthetized with inhaled isoflurane (induction: 3% in oxygen, maintenance: 1.5% in oxygen) and secured in a stereotaxic frame (Kopf, Tujunga, CA). Core body temperature was maintained at ~37°C with a heating pad and eyes were protected with ophthalmic ointment. Lidocaine (1%) was injected under the scalp and an incision was made into the skin at the midline to expose the skull. Using a 27-gauge needle as a scalpel, a small craniotomy (~0.4 mm diameter) was made over the temporal cortex (~4 mm lateral to lambda). A glass micropipette, containing AAV vectors, driven by synapsin promoter for neuron-specific expression, was inserted into the cortex 1 mm past the surface of the skull with a micromanipulator (Kopf). We used AAV9.CaMKII.GCaMP6fast.WPRE.SV40 (GCaMP6f) and AAV9.CAG.Flex.GCaMP6f.WPRE.SV40 for two-photon imaging and AAV9.Syn.GCaMP6slow.WPRE.SV40 (GCaMP6s), AAV9.Syn.GCaMP6f. WPRE.SV40, AAV9.CaMKII.GCaMP6f.WPRE.SV40, and AAV9.CAG.Flex.GCaMP6f.WPRE.SV40 for transcranial imaging experiments (titer $5e^{12}$ – $5e^{13}$ genome copies/mL, Penn Vector Core; [*Chen et al., 2013*]). The glass micropipette was backfilled with mineral oil and connected to a 5 μL glass syringe (Hamilton, Reno, NV). We used a syringe pump (World Precision Instruments, Sarasota, FL) to inject 200–400 nL of this solution over the course of 5 min. The pipette was left in place for 2 min after the end of the injection. The pipette was then removed and the scalp of the mouse was closed with cyanoacrylate adhesive. Mice were fed a diet containing the non-steroidal anti-inflammatory drug carprofen (Medigel, Westbrook, ME) for 24 hr prior to and 48 hr after surgery. Mice were monitored for signs of postoperative stress and pain.

### In vivo imaging

11–24 days after AAV injections, mice were prepared for in vivo calcium imaging. Mice were anesthetized with inhaled isoflurane (induction: 3% in oxygen, maintenance: 1.5% in oxygen) and positioned into a head holder. Core body temperature was maintained at ~37°C with a heating pad and eyes were protected with ophthalmic ointment. Lidocaine (1%) was injected under the scalp and an incision (~1.5 cm long) was made into the skin over the right temporal cortex. The head of the mouse was rotated ~45 degrees in the coronal plane to align the pial surface of the right temporal cortex with the imaging plane of the upright microscope optics. The skull of the mouse was secured to the head holder using dental acrylic (Lang, Wheeling, IL) and cyanoacrylate adhesive. A tube (the barrel of a 25 mL syringe or an SM1 tube from Thorlabs, Newton, NJ) was placed around the animal's body to reduce movement, and the mouse received an injection of the sedative chlorprothixene (0.36 mg/kg intramuscular) to reduce animal movement during in vivo imaging (*Chen et al., 2013*; *Kato et al., 2015*). A dental acrylic reservoir was created to hold warm artificial cerebrospinal fluid (ACSF) over the exposed skull. In preparing the ACSF, we removed contaminating zinc by

incubating with Chelex 100 resin (Biorad, Hercules, CA) for 1 hr. Subsequently, we removed the Chelex by vacuum filtration, and added high purity calcium and magnesium salts (99.995% purity; Sigma-Aldrich, St. Louis, MO). The solution contained in millimolar: 130 NaCl, 3 KCl, 2.4 CaCl$_2$, 1.3 MgCl$_2$, 20 NaHCO$_3$, 3 Hepes, and 10 D-glucose, pH = 7.25–7.35, ~300 mOsm. For better optical access of the auditory cortex, we injected lidocaine-epinephrine (2% lidocaine, 1/100,000 weight/ volume epinephrine) into the temporal muscle and retracted a small portion of the muscle from the skull. Mice were then positioned under the microscope objective in a sound- and light-attenuation chamber containing the microscope and a calibrated speaker (ES1, Tucker-Davis Davis Technologies, Alachua, FL). Acoustic stimuli were calibrated with ¼ inch microphone (Brüel and Kjær, Nærum, Denmark) placed at the location of the animal's ear within the chamber.

## Transcranial imaging

We performed transcranial imaging to locate A1 in each mouse. We removed the isoflurane from the oxygen flowing to the animal and began imaging sound-evoked responses at least ten minutes later (*Issa et al., 2014*). Sounds were delivered from a free-field speaker 10 cm from the left ear of the animal (ES1 speaker, ED1 driver, Tucker-Davis Technologies), controlled by a digital to analog converter with an output rate of 250 kHz (USB-6229, National Instruments, Austin, TX). We used ephus (*Suter et al., 2010*) to generate the sound waveforms and synchronize the sound delivery and image acquisition hardware. We presented 50 or 60 dB SPL, 5 or 6 kHz tones to the animal while illuminating the skull with a blue LED (nominal wavelength of 490 nm, M490L2, Thorlabs). We imaged the change in green GCaMP6 emission with epifluorescence optics (eGFP filter set, U-N41017, Olympus, Center Valley, PA) and a 4X objective (Olympus) using a cooled CCD camera (Rolera, Q-Imaging, Surrey, BC, Canada). Images were acquired at a resolution of 174 × 130 pixels (using 4X spatial binning, each pixel covered an area of = 171.1 µm$^2$ of the image) at a frame rate of 20 Hz. After locating A1 in each animal (see analysis section below), we reanesthetized the mouse with isoflurane and performed a small craniotomy (0.4–1 mm$^2$) adjacent to the location of A1 (the edge of craniotomy was ~0.5 mm medial and ~0.5 mm anterior to A1). Using a micromanipulator (Siskiyou, Grants Pass, OR), we inserted a glass micropipette backfilled with mineral oil and connected to a 5 µL glass syringe into the cortex as above. The pipette contained ACSF including 100 µM of ZX1 and 50 µM Alexa-594. Once the pipette was inserted into the cortex, we removed the isoflurane. After ten minutes of recovery from isoflurane, we presented sound stimuli (12 kHz tones, 30–80 dB SPL, 0.4 s duration, 10 msec ramps) while measuring the changes in GCaMP6 fluorescence. After recording the responses to different sounds (3 to 5 presentations of each sound level), we began to infuse the ZX1 solution into the cortex at a rate of 30 nL/min. We monitored the intracortical spread of the ZX1 solution with red epifluorescence optics (excitation: FF01-543/22, emission: FF01-593/40, dichroic: Di02-R561, Semrock, Rochester, NY) and a green LED (nominal wavelength of 530 nm, M530L2, Thorlabs) for transcranial illumination. After ten to twenty minutes, we observed strong transcranial red fluorescence throughout A1 indicating that the ZX1 solution was present in the cortex. At this point we reduced the pump speed to 9 nL/min and remeasured the sound-evoked responses. For experiments in which we infused 1 mM APV and then 1 mM APV and 100 µM ZX1 into the cortex, we constructed two-barrel infusion pipettes by hot-gluing 2 infusion pipettes together so that their tips were within ~200 µm of each other. Each pipette was backfilled with mineral oil and connected to a 5 µL glass syringe as above. One pipette contained ACSF with APV and 10 µM Alexa-594, and the other contained ACSF with APV, ZX1, and 50 µM Alexa-594. We inserted the tips of both pipettes into the small craniotomy as above. We measured control sound-evoked responses, infused APV and remeasured these responses, and then infused APV and ZX1 and measured these responses a third time. We monitored the diffusion of each solution in the cortex by the increase in red transcranial fluorescence as above. We followed a similar approach for the sequential infusion of vehicle and ZX1 in *Figure 1—figure supplement 1d,e*. Mice were euthanized at the end of the recording session.

## Analysis

To localize A1, we used 50 or 60 dB SPL, 5 or 6 kHz tones and we normalized the sound-evoked change in fluorescence after sound presentation (ΔF) to the baseline fluorescence (F), where F is the average fluorescence of 1 s preceding the sound onset (for each pixel in the movie). We applied a

two-dimensional, low-pass Butterworth filter to each frame of the ΔF/F movie, and then created an image consisting of a temporal average of 10 consecutive frames (0.5 s); the temporal average started at the end of the sound stimulus. This image indicated two sound-responsive regions corresponding to the low frequency tonotopic areas of A1 and the AAF (*Figure 1*). A region of interest (ROI, 150–200 µm x 150–200 µm) over A1 was then used to quantify the sound-evoked responses to 12 kHz sounds. We averaged the fluorescent intensity from all pixels in the ROI for each frame and normalized the ΔF to the F of the ROI to yield ΔF/F responses. ΔF/F responses from 3 to 5 presentations of the same sound level and frequency were averaged. Response amplitude was the peak (50 msec window) of the transcranial response that occurred within one second of the sound onset. To quantify of the effects of ZX1 on the type of gain change (multiplicative/divisive and additive/subtractive) the response amplitudes from each mouse (in control and ZX1) were normalized to the largest response in control. The normalized response to each sound in ZX1 was then plotted against the corresponding control response, and the slope and y-intercept were quantified with a regression line fit through the data.

## Two-photon imaging

For 2-photon imaging in awake mice, we followed the same steps as above to locate A1, but created a larger craniotomy (~1 mm$^2$) over A1 for improved optical access, and inserted the pipette containing ZX1 into the cortex at the edge of this craniotomy. Mode-locked infrared laser light (940 nm, 100–200 mW intensity at the back focal plane of the objective, MaiTai HP, Newport, Santa Clara, CA) was delivered through a galvanometer-based scanning 2-photon microscope (Scientifica, Uckfield, UK) controlled with scanimage (*Pologruto et al., 2003*), using a 40X, 0.8 NA objective (Olympus) with motorized stage and focus controls. We imaged green and red fluorescence simultaneously with 2 photomultiplier tubes using red and green emission filters (FF01-593/40, FF03-525/50, Semrock) and a dichroic splitter (Di02-R561, Semrock). We acquired movies at a frame rate of 5 Hz over an area of 145 µm x 145 µm and at a resolution of 256 × 256 pixels. We imaged neurons in L2/3 at an average depth of 197 µm ± 34 µm from pia, the range represents standard deviation. After identifying A1 L2/3 neurons responding to sounds, we presented different levels (30, 50, 70 dB SPL) of 12 kHz tones (500 msec duration, 20 msec ramps) while monitoring the changes in GCaMP6f fluorescence. We recorded neuronal activity in ten-second long movies and presented sound stimuli 4 s after the start of each movie. We presented different sound stimuli every 30 s. After obtaining movies of responses to different sound stimuli, we began to infuse ZX1. Once ZX1 diffused in A1, we remeasured the responses of the same neurons to the same sounds. Mice were euthanized at the end of the recording session.

## Analysis

To quantify the neuronal responses to sounds we identified neurons that were present in the field of view before and after ZX1 infusion and targeted only these cells for analysis. Using FluoroSNNAP software (*Patel et al., 2015*), we selected ROIs within the soma of each L2/3 neuron from the temporal average of each movie (50 frames, ten-sec long). The pixels in each ROI from each frame were averaged and converted into ΔF/F as above. We then averaged the fluorescent response for 4–7 presentations of the same sound intensity and frequency for each neuron. Sound-evoked responses were measured for 1 s after of the sound onset and were defined as responses if the sound-evoked increases in ΔF/F were larger than the mean +3 standard deviations of the baseline fluorescence measured prior to the sound onset. The response was quantified as the integral of the fluorescence during this 1 s period. For neurons that responded to 12 kHz tones, we quantified the type of gain change (multiplicative/divisive and additive/subtractive) following zinc chelation by plotting the sound-evoked response amplitudes in ZX1 against the corresponding response amplitudes in control conditions. We quantified the slope and y-intercept of this relationship for each neuron with a major axis regression (*Phillips and Hasenstaub, 2016*).

## Histology

To image GCaMP6f expressing neurons in brain slices, mice that had undergone AAV injections were deeply anesthetized with isoflurane and decapitated. Brains were quickly removed and sectioned, with a vibratome (Leica, Buffalo Grove, IL), into 300 µm acute slices containing the temporal

cortex. We prepared the slices in a solution containing (in mM): 2.5 KCl, 1.25 $NaH_2PO_4$, 25 $NaHCO_3$, 0.5 $CaCl_2$, 7, $MgCl_2$, 7 dextrose, 205 sucrose, 1.3 ascorbic acid, and 3 sodium pyruvate (pH 7.35, bubbled with 95% $O_2$/5% $CO_2$). The slices were then transferred and incubated at 36°C in ACSF (see above) for 30 min. Subsequently, slices were transferred to a solution containing 4% paraformaldehyde (Electron Microscopy Sciences, Hattfield, PA) in 0.01M phosphate buffered saline (PBS) and incubated at 4°C overnight. Following this fixation, slices were rinsed five times in PBS and mounted on glass slides. We acquired fluorescent images of GCaMP6f expressing neurons in cortical slices with 2-photon microscopy (see above).

## Experiments with KO mice
Experiments with WT and ZnT3 KO were blinded.

## Statistics
Analysis was performed with MATLAB (Mathworks, Natick, MA) and QuickCalcs (Graphpad, La Jolla, CA). Group data are presented as mean ± standard error of the mean. Pairwise comparisons between groups were performed with the Student's paired t-test, t-test or one sample t-test (for normally distributed data) or the Wilcoxon signed-rank or rank sum tests (for non-normally distributed data). Normality of the distribution of data was assessed with the Lilliefors test. Significance is defined as $p < 0.05$. The sample size for experiments was chosen to be consistent with similar studies in the field, such as in (*Kato et al., 2015*; *Phillips and Hasenstaub, 2016*).

## Acknowledgements
We thank Drs. Elias Aizenman and Brent Doiron for helpful discussions; Drs. Elias Aizenman and John Adelman for comments on the manuscript. We acknowledge Vivek Jayaraman, Ph.D., Douglas S Kim, Ph.D., Loren L Looger, Ph.D., Karel Svoboda, Ph.D. from the GENIE Project, Janelia Research Campus, Howard Hughes Medical Institute for making GCaMP6f and 6s available.

## Additional information

### Funding

| Funder | Grant reference number | Author |
| --- | --- | --- |
| National Institute on Deafness and Other Communication Disorders | R01-DC007905 | Thanos Tzounopoulos |
| National Institute on Deafness and Other Communication Disorders | F32-DC013734 | Charles T Anderson |

The funders had no role in study design, data collection and interpretation, or the decision to submit the work for publication.

### Author contributions
Charles T Anderson, Conceptualization, Formal analysis, Supervision, Investigation, Methodology, Writing—original draft, Writing—review and editing; Manoj Kumar, Conceptualization, Data curation, Formal analysis, Investigation, Methodology, Writing—review and editing; Shanshan Xiong, Data curation, Formal analysis, Investigation, Methodology; Thanos Tzounopoulos, Conceptualization, Data curation, Formal analysis, Supervision, Funding acquisition, Writing—original draft, Project administration, Writing—review and editing

### Author ORCIDs
Charles T Anderson http://orcid.org/0000-0003-3353-0182
Thanos Tzounopoulos http://orcid.org/0000-0003-4583-145X

### Ethics

Animal experimentation: Animals were handled, anesthetized and sacrificed according to methods approved by the University of Pittsburgh Institutional Animal Care and Use Committee. The approved IACUC protocol numbers that were employed for this study were: #14125118 and #14094011.

### Decision letter and Author response

Decision letter https://doi.org/10.7554/eLife.29893.013
Author response https://doi.org/10.7554/eLife.29893.014

## Additional files

### Supplementary files

• Transparent reporting form
DOI: https://doi.org/10.7554/eLife.29893.011

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

## Appendix 1

DOI: https://doi.org/10.7554/eLife.29893.012

# Detailed values for main figures

(*Figure 1f*) 30 dB SPL: control: 4.0 ± 1.6% ΔF/F, ZX1: 2.7 ± 1.9% ΔF/F, p=0.44; 40 dB SPL: control: 3.2 ± 0.7% ΔF/F, ZX1: 4.1 ± 0.3% ΔF/F, p=0.27; 50 dB SPL: control: 6.6 ± 1.2% ΔF/F, ZX1: 7.3 ± 1.2% ΔF/F, p=0.65; 60 dB SPL: control: 6.7 ± 1.1% ΔF/F, ZX1: 7.8 ± 1.2% ΔF/F, p=0.55, 70 dB SPL: control: 7.2 ± 1.1% ΔF/F, ZX1: 16.4 ± 3.5% ΔF/F, p=0.03; 80 dB SPL: control: 8.3 ± 1.4% ΔF/F, ZX1: 19.5 ± 2.7% ΔF/F, p=0.003, paired t-tests, n = 7 mice. Compared to control, vehicle: 50 dB SPL: 6.3 ± 0.6%, p=0.84; 60 dB SPL: 7.0 ± 0.6%, p=0.69; 70 dB SPL: 6.8 ± 0.5%, p=0.99; 80 dB SPL: 7.3 ± 1.3%, p=0.56, n = 3 mice, t-tests.

(*Figure 1g*) Regression fit $R^2$ = 0.33.

(*Figure 1h*) Regression slope in WT = 1.96 ± 0.49 (vs 1, p=0.007, n = 7 mice, one sample t-test).

(*Figure 1i*) Regression y-intercept in WT = −0.12 ± 0.36 (vs 0, p=0.75, n = 7 mice, one sample t-test).

(*Figure 1j*) 30 dB SPL: control: 4.0 ± 0.6% ΔF/F, ZX1: 4.6 ± 0.7% ΔF/F, p=0.54; 40 dB SPL: control: 5.8 ± 3.9% ΔF/F, ZX1: 7.0 ± 3.6% ΔF/F, p=0.72; 50 dB SPL: control: 8.7 ± 2.9% ΔF/F, ZX1: 9.5 ± 4.5% ΔF/F, p=0.84; 60 dB SPL: control: 9.3 ± 2.7% ΔF/F, ZX1: 7.7 ± 2.0% ΔF/F, p=0.62; 70 dB SPL: control: 15.3 ± 5.3% ΔF/F, ZX1: 13.3 ± 3.0% ΔF/F, p=0.79; 80 dB SPL: baseline: 21.3 ± 5.3% ΔF/F, ZX1: 17.8 ± 4.6% ΔF/F, p=0.72, n = 6 mice, paired t-tests.

(*Figure 1k*) Regression fit $R^2$ = 0.66.

(*Figure 1l*) Regression slope in ZnT3 KO = 0.78 ± 0.11 (vs. 1, p=0.10, n = 6 mice, one sample t-test).

(*Figure 1m*) Regression y-intercept in ZnT3 KO = 0.09 ± 0.07 (vs. 0, p=0.23, n = 6 mice, one sample t-test).

(*Figure 2b*) 30 dB SPL: control: 1.9 ± 0.5% ΔF/F, ZX1: 2.2 ± 0.9% ΔF/F, p=0.53; 40 dB SPL: control: 3.0 ± 0.6% ΔF/F, ZX1: 2.4 ± 0.9% ΔF/F, p=0.19; 50 dB SPL: control: 5.8 ± 0.9% ΔF/F, ZX1: 5.1 ± 0.4% ΔF/F, p=0.48; 60 dB SPL: control: 8.9 ± 1.5% ΔF/F, ZX1: 6.7 ± 1.0% ΔF/F, p=0.02, 70 dB SPL: control: 12.2 ± 1.4% ΔF/F, ZX1: 8.3 ± 1.2% ΔF/F, p=0.001; 80 dB SPL: control: 11.4 ± 1.1% ΔF/F, ZX1: 8.8 ± 1.0% ΔF/F, p=0.003, paired t-tests, n = 10 mice.

(*Figure 2c*) Regression fit $R^2$ = 0.66. Regression slope = 0.72 ± 0.06 (vs. 1, p=0.001, n = 10 mice, one sample t-test). Regression y-intercept = 0.05 ± 0.04 (vs. 0, p=0.32, n = 10 mice, one sample t-test).

(*Figure 2e*) 30 dB SPL: control: 3.9 ± 1.3% ΔF/F, ZX1: 5.8 ± 1.7% ΔF/F, p=0.009; 40 dB SPL: control: 3.9 ± 1.0% ΔF/F, ZX1: 7.3 ± 1.9% ΔF/F, p=0.04; 50 dB SPL: control: 4.7 ± 1.2% ΔF/F, ZX1: 9.5 ± 2.5% ΔF/F, p=0.04; 60 dB SPL: control: 5.2 ± 1.3% ΔF/F, ZX1: 14.5 ± 3.4% ΔF/F, p=0.006, 70 dB SPL: control: 5.2 ± 1.4% ΔF/F, ZX1: 15.7 ± 3.7% ΔF/F, p=0.02; 80 dB SPL: control: 5.3 ± 1.3% ΔF/F, ZX1: 16.0 ± 3.6% ΔF/F, p=0.01, paired t-tests, n = 4 mice.

(*Figure 2f*) Regression fit $R^2$ = 0.37. Regression slope = 3.2 ± 0.86 (vs. 1, p=0.03, n = 4 mice, one sample t-test). Regression y-intercept = 0.05 ± 0.28 (vs. 0, p=0.385, n = 4 mice, one sample t-test).

(*Figure 2h*) 30 dB SPL: control: 1.0 ± 0.3% ΔF/F, ZX1: 1.8 ± 0.4% ΔF/F, p=0.01; 40 dB SPL: control: 1.9 ± 0.4% ΔF/F, ZX1: 3.8 ± 1.1% ΔF/F, p=0.04; 50 dB SPL: control: 3.4 ± 1.2% ΔF/F, ZX1: 6.3 ± 1.9% ΔF/F, p=0.03; 60 dB SPL: control: 4.3 ± 1.5% ΔF/F, ZX1: 7.3 ± 2.1% ΔF/F, p=0.03, 70 dB SPL: control: 7.9 ± 2.3% ΔF/F, ZX1: 8.1 ± 2.6% ΔF/F, p=0.97; 80 dB SPL: control: 12.1 ± 4.0% ΔF/F, ZX1: 14.8 ± 5.4% ΔF/F, p=0.19, paired t-tests, n = 4 mice.

(*Figure 2i*) Regression fit $R^2$ = 0.87. Regression slope = 0.97 ± 0.09 (vs. 1, p=0.77, n = 4 mice, one sample t-test). Regression y-intercept = 0.17 ± 0.05 (vs. 0, p=0.01, n = 4 mice, one sample t-test).

(*Figure 2k*) 30 dB SPL: control: 0.7 ± 0.1% ΔF/F, ZX1: 0.8 ± 0.1% ΔF/F, p=0.62; 40 dB SPL: control: 0.8 ± 0.2% ΔF/F, ZX1: 0.8 ± 0.04% ΔF/F, p=0.81; 50 dB SPL: control: 1.3 ± 0.2% ΔF/F, ZX1: 1.5 ± 0.1% ΔF/F, p=0.00.16; 60 dB SPL: control: 1.5 ± 0.2% ΔF/F, ZX1: 2.1 ± 0.2% ΔF/F,

p=0.02, 70 dB SPL: control: 2.1 ± 0.4% ΔF/F, ZX1: 3.3 ± 0.9% ΔF/F, p=0.01; 80 dB SPL: control: 2.3 ± 0.6% ΔF/F, ZX1: 3.5 ± 1.0% ΔF/F, p=0.04, paired t-tests, n = 4 mice.

(*Figure 2l*) Regression fit $R^2$ = 0.53. Regression slope = 1.21 ± 0.76 (vs. 1, p=0.38, n = 4 mice, one sample t-test). Regression y-intercept = 0.09 ± 0.18 (vs. 0, p=0.62, n = 4 mice, one sample t-test).

(*Figure 3b*) 30 dB SPL: in APV: 3.5 ± 0.6% ΔF/F, ZX1: 4.1 ± 1.1% ΔF/F, p=0.38; 40 dB SPL: in APV: 4.7 ± 1.1% ΔF/F, ZX1: 5.3 ± 1.3% ΔF/F, p=0.19; 50 dB SPL: in APV: 4.3 ± 1.0% ΔF/F, ZX1: 4.7 ± 0.9% ΔF/F, p=0.28; 60 dB SPL: in APV: 5.13 ± 1.5% ΔF/F, ZX1: 5.1 ± 1.3% ΔF/F, p=0.99, 70 dB SPL: in APV: 6.5 ± 1.4% ΔF/F, ZX1: 9.6 ± 2.4% ΔF/F, p=0.04; 80 dB SPL: in APV: 8.5 ± 2.0% ΔF/F, ZX1: 11.1 ± 2.5% ΔF/F, p=0.009, paired t-tests, n = 6 mice.

(*Figure 3c*) 30 dB SPL: in APV: 1.5 ± 0.1% ΔF/F, ZX1: 5.4 ± 1.2% ΔF/F, p=0.04; 40 dB SPL: in APV: 2.6 ± 0.6% ΔF/F, ZX1: 7.0 ± 1.3% ΔF/F, p=0.03; 50 dB SPL: in APV: 2.2 ± 0.6% ΔF/F, ZX1: 5.7 ± 0.7% ΔF/F, p=0.02; 60 dB SPL: in APV: 3.6 ± 0.7% ΔF/F, ZX1: 9.5 ± 2.5% ΔF/F, p=0.04, 70 dB SPL: in APV: 5.3 ± 0.9% ΔF/F, ZX1: 13.6 ± 2.6% ΔF/F, p=0.01; 80 dB SPL: in APV: 5.8 ± 0.8% ΔF/F, ZX1: 15.3 ± 3.0% ΔF/F, p=0.01, paired t-tests, n = 6 mice.

(*Figure 3e*) 30 dB SPL: in APV: 2.2 ± 0.6% ΔF/F, ZX1: 1.2 ± 0.2% ΔF/F, p=0.20; 40 dB SPL: in APV: 2.9 ± 0.8% ΔF/F, ZX1: 1.3 ± 0.2% ΔF/F, p=0.17; 50 dB SPL: in APV: 3.6 ± 0.6% ΔF/F, ZX1: 2.4 ± 0.6% ΔF/F, p=0.31; 60 dB SPL: in APV: 4.2 ± 1.0% ΔF/F, ZX1: 2.6 ± 0.4% ΔF/F, p=0.25, 70 dB SPL: in APV: 6.2 ± 1.2% ΔF/F, ZX1: 4.6 ± 1.4% ΔF/F, p=0.12; 80 dB SPL: in APV: 10.4 ± 4.2% ΔF/F, ZX1: 7.7 ± 2.5% ΔF/F, p=0.21, paired t-tests, n = 4 mice.

(*Figure 3f*) 30 dB SPL: in APV: 8.4 ± 3.2% ΔF/F, ZX1: 8.4 ± 2.9% ΔF/F, p=0.98; 40 dB SPL: in APV: 11.4 ± 5.3% ΔF/F, ZX1: 9.1 ± 3.2% ΔF/F, p=0.44; 50 dB SPL: in APV: 17.0 ± 6.2% ΔF/F, ZX1: 17.1 ± 5.4% ΔF/F, p=0.97; 60 dB SPL: in APV: 23.2 ± 7.4% ΔF/F, ZX1: 21.3 ± 6.8% ΔF/F, p=0.10, 70 dB SPL: in APV: 23.9 ± 7.4% ΔF/F, ZX1: 24.5 ± 7.8% ΔF/F, p=0.31; 80 dB SPL: in APV: 24.4 ± 7.6% ΔF/F, ZX1: 23.7 ± 7.5% ΔF/F, p=0.33, paired t-tests, n = 4 mice.

(*Figure 4c*) 30 dB SPL: control: 17.5 ± 5.1% ΔF/F, ZX1: 19.5 ± 2.7% ΔF/F, p=0.92; 50 dB SPL: control: 30.1 ± 5.3% ΔF/F, ZX1: 30.4 ± 5.0% ΔF/F, p=0.11; 70 dB SPL: control: 42.6 ± 10.1% ΔF/F, ZX1: 26.7 ± 4.6% ΔF/F, p=0.04, signed-rank tests, n = 86 neurons from 8 mice.

(*Figure 4d*) Regression slope = 0.49 ± 0.14 (vs. 1, p=0.001, one sample t-test, n = 86 neurons from n = 8 mice). Regression y-intercept = −0.04 ± 0.04 (vs. 0, p=0.32, one-sample t-test, n = 86 neurons from n = 8 mice).

(*Figure 4g*) 30 dB SPL: control: 19.4 ± 4.8% ΔF/F, ZX1: 19.7 ± 3.7% ΔF/F, p=0.69; 50 dB SPL: control: 55.9 ± 8.1% ΔF/F, ZX1: 83.9 ± 14.7% ΔF/F, p=0.016; 70 dB SPL: control: 68.9 ± 9.5% ΔF/F, ZX1: 150.6 ± 25.5% ΔF/F, p=0.001, signed-rank tests, n = 40 neurons from 6 mice.

(*Figure 4h*) Regression slope = 2.00 ± 0.47 (vs. 1, p=0.04, one sample t-test, n = 40 neurons from 8 mice). Regression y-intercept = −0.07 ± 0.22 (vs. 0, p=0.75, one-sample t-test, n = 40 neurons from 6 mice).

(*Figure 5c*) 30 dB SPL: control: −4.6 ± 5.0% ΔF/F, ZX1: 27.8 ± 0.9% ΔF/F, p=0.003; 50 dB SPL: control: 13.6 ± 10.1% ΔF/F, ZX1: 53.4 ± 11.5% ΔF/F, p=0.02; 70 dB SPL: control: 60.1 ± 21.0% ΔF/F, ZX1: 82.8 ± 25.2% ΔF/F, p=0.38, signed-rank tests, n = 12 neurons from 4 mice.

(*Figure 5d*) Regression slope = 1.18 ± 0.45 (vs. 1, p=0.70, one sample t-test, n = 12 neurons from 4 mice). Regression y-intercept = 24.1 ± 13.3 (vs. 0, p=0.001, one-sample t-test, n = 12 neurons from 4 mice).

(*Figure 5g*) 30 dB SPL: control: 5.8 ± 5.7% ΔF/F, ZX1: 19.6 ± 7.6% ΔF/F, p=0.17; 50 dB SPL: control: 22.1 ± 7.2% ΔF/F, ZX1: 66.2 ± 36.1% ΔF/F, p=0.28; 70 dB SPL: control: 149.6 ± 36.3% ΔF/F, ZX1: 129.1 ± 46.6% ΔF/F, p=0.39, signed-rank tests, n = 30 neurons from 6 mice.

(*Figure 5h*) Regression slope = 0.91 ± 0.25 (vs. 1, p=0.74, one sample t-test, n = 30 neurons from 6 mice). Regression y-intercept = 0.10 ± 0.09 (vs. 0, p=0.29, one-sample t-test, n = 30 neurons from 6 mice).

