## [Decision Letter]

[Editors’ note: a previous version of this study was rejected after peer review, but the authors submitted for reconsideration. The first decision letter after peer review is shown below.]

Thank you for submitting your work for consideration by *eLife*. Your article has been reviewed by two peer reviewers, and the evaluation has been overseen by a Reviewing Editor and a Senior Editor. The reviewers have opted to remain anonymous.

Our decision has been reached after consultation between the reviewers. Based on these discussions and the individual reviews below, we regret to inform you that your work will not be considered further for publication in *eLife*.

Overall, the reviewers felt that this is an intriguing study and were convinced that you had shown an effect of zinc chelation on auditory processing. They were also impressed by the range of approaches used and stated that these findings are likely to inspire follow up work in this field. The reviewers did, however, raise a large number of issues, with several requests for additional data. Reviewer #2 highlighted the lack of mechanistic insights provided by the current data, while reviewer #3 was concerned that the effects of zinc removal are restricted to loud sounds and that the data are too variable to make firm conclusions about gain control. Both reviewers pointed out that it is not clear that (or at least to what extent) the effects described necessarily indicate an involvement of synaptically released zinc. While recognizing the amount of work that has already gone into this study and that it is obviously not possible to cover everything in one article, it was clear from the subsequent discussion between the reviewers that the study currently contains many loose ends and that these could not easily be fixed in a revision. Unfortunately, we have therefore concluded that we will not be able to consider your paper further at *eLife*.

Reviewer #2:

This paper shows interesting effects of zinc chelation – or removal of synaptically released zinc via genetic deletion of the vesicular zinc transporter ZnT3 – on auditory processing in mice. Disrupting zinc signaling is shown to have effects at different levels: 1) apparently (see below) knocking out ZnT3; 2) Responses to loud (>70 dB) sound stimuli in A1 neuronal populations is increased by ZX1 or ZnT3 deletion.

These results show quite clearly that zinc modulates elements involved in auditory processing. A significant strength of this paper is that effects are shown in several different experimental paradigms that examine auditory signaling at several levels. Clearly, this laboratory is poised to examine how synaptic mechanisms influence auditory processing in physiologically and behaviorally relevant ways. A significant weakness of the paper is that little insight is provided into the actual synaptic mechanisms within the auditory pathway that underlie the observed effects of zinc. The authors claim at the beginning of the Discussion that their "results provide fundamental synaptic mechanisms governing normal neural processing." I think this is a significant overstatement. The results don't even show directly that the primary effects of zinc are actually at synapses (although, given previous work from this and other labs, this certainly seems likely). If we assume that the primary effects occur at excitatory synapses, we don't know whether these synapses are on principal neurons, interneurons, or both, and whether the effects reflect primarily modulation of AMPA or NMDA receptors.

The authors recognize these gaps and make the reasonable point that further experiments are required to elucidate the underlying mechanisms (Discussion paragraph six). I think that the experiments presented here provide ample motivation for those future experiments, and I accept the implication that the more mechanistically detailed experiments are beyond the scope of the present paper. I would, however, ask the authors to tone down the language regarding insights into fundamental mechanisms.

Reviewer #3:

General Comments: I think the biggest problem I have with the results is that it is very puzzling that the effects of synaptic zinc are restricted to loud sounds > 70 dB – And the authors seem to try to explain this odd and selective effect to loud sounds with a lot of cursory commentary. Finally I feel that the authors extrapolate too much from these limited results to autism and schizophrenia in their Discussion.

A bit more detail:

1) Figure 1 not entirely convincing – so much variability in responses to the 12 kHz tone in control and ZX1 conditions. The fact that the regression line is above unity is not a persuasive argument for divisive gain control.

2) The authors show that effects for 70 and 80 dB in Figure 1, but in 2C only get significant effects at 80 dB!

3) All results in normal mice are based on inference from the effects of ZX1 – which chelates and thus reduces synaptically evoked zinc, but presumably also reduces extracellular Zn as well. The authors should provide clearer evidence of the relative effects on synaptically evoked Zn and extracellular Zn level with careful measurements. It would also be helpful to have a more direct technique to specifically enhance the concentration of Zn in synaptic transmitter vesicles. Would additional synaptic evoked zinc release lead to even more inhibition of A1 responses to loud sounds? Or to effects on a broader range of intensities? Are there mouse mutants with enhanced Zn concentration in synaptic vesicles?

4) What about layer II, III cells with non-monotonic intensity tuning? Do they also show the effects of ZXI as cells with monotonic intensity tuning?

5) The authors focused on layer II, III cells – what about layer IV cells? Are the effects of Zn at loud intensities present only for the supragranular cells or also observed in layer IV cells, which presumably are activated primarily from the thalamic inputs?

6) In general, one wonders how important ethologically these Zn effects would be for mice – given that most sounds in their normal acoustic environment are less than 70 dB.

7) What is the clinical evidence that schizophrenics are selectively impaired in sound perception at loud intensities of sound? My impression is that schizophrenics are often hypersensitive to all sounds, including faint ones.

Obviously a lot of work went into this study. The findings are intriguing, but the story feels incomplete.

[Editors’ note: what now follows is the decision letter after the authors submitted for further consideration.]

Thank you for submitting your article "Cell-specific gain modulation by synaptically released zinc in cortical circuits of audition" for consideration by *eLife*. Your article has been reviewed by three peer reviewers, and the evaluation has been overseen by a Reviewing Editor and Andrew King as the Senior Editor. The following individuals involved in review of your submission have agreed to reveal their identity: Richard Dyck (Reviewer #2).

The reviewers have discussed the reviews with one another and the Reviewing Editor has drafted this decision to help you prepare a revised submission.

Summary:

The reviewers agreed that this study makes an important and substantial contribution to elucidating the possible role of synaptically released zinc in the normal function of auditory cortex, and that it is likely to inspire future studies in this area. Most of the existing literature has focused on the involvement of synaptic zinc in relation to pathological conditions. By showing that synaptic zinc can change the gain of sound-evoked responses in a neuron-specific fashion, this work provides clear evidence for a contribution to normal brain function. It is therefore likely to have a significant impact on the field of zinc neurobiology as a whole.

Essential revisions:

The reviewers agreed that the manuscript is, generally, clearly written and logically organized, and substantially improved relative to the previous version. The following issues were raised:

1) There was some discussion among the reviewers about the selectivity and specificity of ZX1 binding. While it was agreed, on the basis of previous work, that the effects of ZX1 are unlikely to be caused by chelating extracellular magnesium or calcium, this issue should be addressed in the manuscript, particularly as two-photon calcium imaging was used to measure the activity of cortical neurons.

2) The authors assume that, at high sound intensities, zinc inhibition of PV interneurons leads to potentiation of the principal neurons, and this effect is NMDA-dependent, because this relationship is not seen when an NMDA antagonist is applied. Please comment on the NMDA-dependent mechanism by which the PV interneurons are normally inhibiting the principal neurons.

3) The bar graphs showing average change in fluorescence per dB should be removed, since the information they provide is either redundant or confusing. For example, the bar graph in Figure 1 shows a 5% change/10dB, whereas the data in Figure 1 show that there is absolutely no change between 30 and 60 dB, and that all the change is accounted for at 70 and 80 dB. The data depicted in Figure 1 are redundant. In other cases (Figure 2 and Figure 4), the bar graphs either simply confirm or actually contradict the data in the corresponding line graphs. This is also true of Figure 1—figure supplement 1, where only show the bar graph, but not the potentially interesting intensity-related data, are included.

4) The authors should make it clear that their data are specific to circuits in layer 2/3 of auditory cortex. Considering the differential innervation of zincergic inputs in different layers, together with the differential distribution of PV (mostly layers 2/3, some 4), SOM (mostly layers 4, 5 and 6) and VIP (fewer than the other two populations, but mostly located superficially) interneurons in auditory cortex, as well as unique within-column and inter-columnar information processing circuits, their results may not apply to the deeper layers of the cortex.

---

## [Author Response]

[Editors’ note: the author responses to the first round of peer review follow.]

Overall, the reviewers felt that this is an intriguing study and were convinced that you had shown an effect of zinc chelation on auditory processing. They were also impressed by the range of approaches used and stated that these findings are likely to inspire follow up work in this field. The reviewers did, however, raise a large number of issues, with several requests for additional data. Reviewer #2 highlighted the lack of mechanistic insights provided by the current data, while reviewer #3 was concerned that the effects of zinc removal are restricted to loud sounds and that the data are too variable to make firm conclusions about gain control. Both reviewers pointed out that it is not clear that (or at least to what extent) the effects described necessarily indicate an involvement of synaptically released zinc. While recognizing the amount of work that has already gone into this study and that it is obviously not possible to cover everything in one article, it was clear from the subsequent discussion between the reviewers that the study currently contains many loose ends and that these could not easily be fixed in a revision. Unfortunately, we have therefore concluded that we will not be able to consider your paper further at eLife.

1) To address the lack of mechanistic insights we collected and analyzed new data to uncover the cellular and molecular mechanisms underlying zinc-mediated gain modulation in the auditory cortex. Namely, we performed new experiments to investigate the effect of synaptic zinc on the gain of principal neurons, as well as interneurons (PV, SOM, VIP interneurons; Figure 2, Figure 4 and Figure 5); and we investigated the role of NMDA and nonNMDA receptors on the zinc–mediated modulation (Figure 3). Our two-photon and transcranial experiments support the cell-specific effect of synaptic zinc on A1 neurons: synaptic zinc increases the gain of sound-evoked in principal neurons, but decreases the gain of PV and SOM interneurons. These effects are sound intensity dependent. Moreover, our results highlight that synaptic zinc targets non-NMDARs to modulate the gain of principal and PV neurons; and suggest that NMDAR-mediated signaling may also be involved in the zinc-mediated gain control of SOM and VIP neurons. We therefore provide cellular and molecular mechanistic insight in our study.

2) Regarding the issue that the effects of zinc removal are restricted to loud sounds, our new experiments and analyses now show effects of zinc at low (30 dB SPL) and high sound intensities (70-80 dB SPL; Figure 2–Figure 5). These effects highlight the ethological relevance of zinc signaling in modulating cortical sound-evoked responses.

3) Regarding the issue of providing firm data for gain control, Figure 1, current Figure 1, contained (and contains) population data from all neuronal types in A1. Therefore, the observed variability may reflect the heterogeneous response of these neurons to sound. To address this issue, we performed new experiments and analyses to investigate how synaptic zinc affects the gain of populations of specific classes of auditory cortical neurons, such as principal, PV, SOM and VIP neurons (Figure 2). Moreover, we performed new experiments and analyses on principal, PV, SOM and VIP neurons with two-photon imaging (Figure 4 and Figure 5). Our new results are less variable and firmly support zinc-mediated gain modulation of the different neuronal types.

4) To address the issue of the extent of the involvement of the synaptically released zinc, we performed additional analysis of our results, as suggested by reviewer 2. To track the origin of the extracellular zinc modulating A1 gain control, we studied A1 gain control in ZnT3 KO mice, which lack the vesicular zinc transporter ZnT3 and synaptic zinc (Cole et al., 1999). Compared to WT mice, ZnT3 KO mice showed increased change of the neuronal response per 10 dB SPL (Figure 1—figure supplement 1), suggesting that synaptic zinc inhibits the gain of A1 responses. Moreover, zinc chelation in KO mice had no effect on the gain of A1 (Figure 1, Materials and methods), suggesting that, in WT mice, ZX1 enhances the A1 gain via chelation of extracellular, ZnT3-dependent zinc. Because extracellular tonic, not synaptically-evoked, zinc levels in brain slices are ZnT3-independent (Anderson et al., 2015), our results suggest that synaptically released zinc modulates gain control of sound-evoked responses in A1.

Additionally, to track the origin of the extracellular zinc increasing the gain of principal neurons, we performed new experiments to study the effects of ZX1 in ZnT3 KO mice, which had been injected with AAV-GCaMP6fCaMKII. Zinc chelation had no effect on the sound-evoked responses of principal neurons in ZnT3 KO mice, indicating that synaptic zinc increases the gain of principal neurons (Figure 2—figure supplement 1).

Based on the editorial and reviewer’s comments, we opted to focus our manuscript on the role of synaptic zinc on the gain control of A1 principal neurons and interneurons. Our results highlight synaptic zinc as a novel modulator of gain control in the auditory cortex, and provide insights on the cellular and molecular mechanisms underlying this modulation.

The effects of zinc on the frequency tuning of A1 principal neurons and interneurons; as well as the effects of zinc on frequency discrimination will be the focus of our next manuscript.

Reviewer #2:[…] These results show quite clearly that zinc modulates elements involved in auditory processing. A significant strength of this paper is that effects are shown in several different experimental paradigms that examine auditory signaling at several levels. Clearly, this laboratory is poised to examine how synaptic mechanisms influence auditory processing in physiologically and behaviorally relevant ways. A significant weakness of the paper is that little insight is provided into the actual synaptic mechanisms within the auditory pathway that underlie the observed effects of zinc. The authors claim at the beginning of the Discussion that their "results provide fundamental synaptic mechanisms governing normal neural processing." I think this is a significant overstatement. The results don't even show directly that the primary effects of zinc are actually at synapses (although, given previous work from this and other labs, this certainly seems likely). If we assume that the primary effects occur at excitatory synapses, we don't know whether these synapses are on principal neurons, interneurons, or both, and whether the effects reflect primarily modulation of AMPA or NMDA receptors.

1) To address the lack of mechanistic insights, we performed new experiments and analyses that provide insight into the cellular and molecular mechanisms underlying the observed effects of zinc (Figure 2–Figure 5). For more details: see point-by-point response to the Editors’ comments, responses 1, 2 and 4.

2) We agree with the reviewer’s comment on the “overstatements”, and we modified the Discussion accordingly. Moreover, we removed the overstatements throughout the manuscript. Given the new mechanistic experiments, we now know that the zinc-mediated modulatory effects occur at principal neurons and interneurons; and non-NMDA receptors, as well as NMDA receptors, are involved in this modulation

The authors recognize these gaps and make the reasonable point that further experiments are required to elucidate the underlying mechanisms (Discussion paragraph six). I think that the experiments presented here provide ample motivation for those future experiments, and I accept the implication that the more mechanistically detailed experiments are beyond the scope of the present paper. I would, however, ask the authors to tone down the language regarding insights into fundamental mechanisms.

We agree with the reviewer and we removed and/or toned down the language throughout the manuscript, regarding into fundamental mechanisms.

Reviewer #3:I think the biggest problem I have with the results is that it is very puzzling that the effects of synaptic zinc are restricted to loud sounds > 70 dB – And the authors seem to try to explain this odd and selective effect to loud sounds with a lot of cursory commentary. Finally I feel that the authors extrapolate too much from these limited results to autism and schizophrenia in their Discussion.

1) Our new experiments and analyses (Figure 2–Figure 5) now show effects of synaptic zinc at low (30 dB SPL) and high sound intensities (70-80 dB SPL). These effects highlight the ethological relevance of synaptic zinc signaling in modulating cortical responses to sound. For more details: see point-by-point response to the Editors’ comments, responses 2 (and 1).

2) We agree with the reviewer about autism and schizophrenia and we removed these topics in the revised Discussion.

1) Figure 1 not entirely convincing – so much variability in responses to the 12 kHz tone in control and ZX1 conditions. The fact that the regression line is above unity is not a persuasive argument for divisive gain control.

See point-by-point response to the Editors’ comments, response 3

2) The authors show that effects for 70 and 80 dB in Figure 1, but in 2C only get significant effects at 80 dB!

Due to the focus of our revised manuscript on the role of synaptic zinc on the gain control of A1 principal neurons and interneurons, we removed Figure 2 from the manuscript, which was focused on frequency tuning.

3) All results in normal mice are based on inference from the effects of ZX1 – which chelates and thus reduces synaptically evoked zinc, but presumably also reduces extracellular Zn as well. The authors should provide clearer evidence of the relative effects on synaptically evoked Zn and extracellular Zn level with careful measurements.

See point-by-point response to the Editors’ comments, response 4.

It would also be helpful to have a more direct technique to specifically enhance the concentration of Zn in synaptic transmitter vesicles. Would additional synaptic evoked zinc release lead to even more inhibition of A1 responses to loud sounds? Or to effects on a broader range of intensities? Are there mouse mutants with enhanced Zn concentration in synaptic vesicles?

This is an important point and we are very interested in investigating how increased levels of synaptic zinc affect the sound-evoked responses of A1 neurons. Experiments addressing this issue are outside the scope of this manuscript and will require new tools, but will be pursued in future studies in our lab.

4) What about layer II, III cells with non-monotonic intensity tuning? Do they also show the effects of ZXI as cells with monotonic intensity tuning?

This comment was not very clear to us. Based on our understanding, we assume that the reviewer asks what is the effect of ZX1 on the frequency tuning of layer II, III cells with monotonic and non-monotonic frequency tuning. Due to the focus of our revised manuscript on the role of synaptic zinc on the gain control of A1 principal neurons and interneurons, we removed all the graphs from the manuscript that were focused on frequency tuning.

5) The authors focused on layer II, III cells – what about layer IV cells? Are the effects of Zn at loud intensities present only for the supragranular cells or also observed in layer IV cells, which presumably are activated primarily from the thalamic inputs?

In the revised manuscript we added new experiments and analyses showing that the effects of synaptic zinc on cortical gain are cell-type specific within layer 2/3. We are currently investigating the effects of zinc on different cell-types in L4 and L5. These experiments are outside the scope of this manuscript, but will be pursued in future studies in our lab.

6) In general, one wonders how important ethologically these Zn effects would be for mice – given that most sounds in their normal acoustic environment are less than 70 dB.

In our new experiments and analyses addressing the effect of zinc on the sound-evoked responses of different cell types (Figure 2, Figure 4 and Figure 5), we now show effects of synaptic zinc at low (30 dB SPL) and high sound intensities (70-80 dB SPL). These effects reflect the ethological relevance of synaptic zinc signaling in modulating cortical responses to sound. Moreover, our results support that synaptic zinc is an endogenous modulator that fine tunes cortical gain in principal neurons and interneurons.

7) What is the clinical evidence that schizophrenics are selectively impaired in sound perception at loud intensities of sound? My impression is that schizophrenics are often hypersensitive to all sounds, including faint ones.

Due to the focus of our revised manuscript on the role of synaptic zinc on the gain control of A1 principal neurons and interneurons, we removed from our discussion the topics of schizophrenia and autism.

Obviously a lot of work went into this study. The findings are intriguing, but the story feels incomplete.

We performed many additional experiments and analyses; we included several new figures and analyses (Figure 2–Figure 5) and edited the text (including the title) according to the editorial and reviewers’ comments. We hope that these changes address both the editorial letter and spirit of the reviews and that the story of cortical gain modulation by synaptic zinc feels now complete.

[Editors' note: the author responses to the re-review follow.]

1) There was some discussion among the reviewers about the selectivity and specificity of ZX1 binding. While it was agreed, on the basis of previous work, that the effects of ZX1 are unlikely to be caused by chelating extracellular magnesium or calcium, this issue should be addressed in the manuscript, particularly as two-photon calcium imaging was used to measure the activity of cortical neurons.

To address this concern, we included this paragraph in the revised manuscript (Results section):

“Consistent with the high selectivity of ZX1 over calcium and other biologically relevant metal ions such as magnesium (Pan et al., 2011), the lack of ZX1 effects on the sound-evoked responses in ZnT3 KO mice further validate that the observed effects of ZX1 on the sound evoked responses of WT mice are due to the chelation of synaptic zinc – and not to the potential non-specific effects of ZX1 on calcium or other metal ions.”

2) The authors assume that, at high sound intensities, zinc inhibition of PV interneurons leads to potentiation of the principal neurons, and this effect is NMDA-dependent, because this relationship is not seen when an NMDA antagonist is applied. Please comment on the NMDA-dependent mechanism by which the PV interneurons are normally inhibiting the principal neurons.

To address this concern, we included this paragraph in the revised manuscript (Discussion section):

“Blockade of NMDARs with APV reduced the sound-evoked responses of PV neurons (Figure 3—figure supplement 1) and increased the sound-evoked responses of principal neurons (Figure 3—figure supplement 1). […]This suggests that NMDAR signaling may serve to increase the sound evoked-responses of PV neurons via a combination of direct excitation of PV neurons and reduced VIP-mediated sound-evoked inhibition to these cells. Future studies will be required to explore these possibilities.”

3) The bar graphs showing average change in fluorescence per dB should be removed, since the information they provide is either redundant or confusing. For example, the bar graph in Figure 1 shows a 5% change/10dB, whereas the data in Figure 1 show that there is absolutely no change between 30 and 60 dB, and that all the change is accounted for at 70 and 80 dB. The data depicted in Figure 1 are redundant. In other cases (Figure 2 and Figure 4), the bar graphs either simply confirm or actually contradict the data in the corresponding line graphs. This is also true of Figure 1—figure supplement 1, where only show the bar graph, but not the potentially interesting intensity-related data, are included.

To address these concerns, in our revised figures we removed these bar graphs from Figure 1, Figure 2, Figure 4 and 5. Moreover, we replaced the bar graph in Figure 1—figure supplement 1 with an intensity-related comparison between WT and ZnT3 KO.

4) The authors should make it clear that their data are specific to circuits in layer 2/3 of auditory cortex. Considering the differential innervation of zincergic inputs in different layers, together with the differential distribution of PV (mostly layers 2/3, some 4), SOM (mostly layers 4, 5 and 6) and VIP (fewer than the other two populations, but mostly located superficially) interneurons in auditory cortex, as well as unique within-column and inter-columnar information processing circuits, their results may not apply to the deeper layers of the cortex.

To address this concern, we included this paragraph in the revised manuscript

(Discussion section):

“Our findings are mostly specific to cells and circuits in L2/3. […] Future studies will be required to address the effects of synaptic zinc on the gain of neurons and circuits located outside of L2/3.”